# High social status males experience accelerated epigenetic aging in wild baboons

Jordan A Anderson[1†], Rachel A Johnston[1†], Amanda J Lea[2,3,4], Fernando A Campos[2,5], Tawni N Voyles[1], Mercy Y Akinyi[6], Susan C Alberts[1,2], Elizabeth A Archie[7], Jenny Tung[1,2,8,9]*

[1]Department of Evolutionary Anthropology, Duke University, Durham, United States; [2]Department of Biology, Duke University, Durham, United States; [3]Lewis-Sigler Institute for Integrative Genomics, Carl Icahn Laboratory, Princeton University, Princeton, United States; [4]Department of Ecology and Evolution, Princeton University, Princeton, United States; [5]Department of Anthropology, University of Texas at San Antonio, San Antonio, United States; [6]Institute of Primate Research, National Museums of Kenya, Nairobi, Kenya; [7]Department of Biological Sciences, University of Notre Dame, Notre Dame, United States; [8]Duke Population Research Institute, Duke University, Durham, United States; [9]Canadian Institute for Advanced Research, Toronto, Canada

*For correspondence:
jenny.tung@duke.edu

[†]These authors contributed equally to this work

**Abstract** Aging, for virtually all life, is inescapable. However, within populations, biological aging rates vary. Understanding sources of variation in this process is central to understanding the biodemography of natural populations. We constructed a DNA methylation-based age predictor for an intensively studied wild baboon population in Kenya. Consistent with findings in humans, the resulting 'epigenetic clock' closely tracks chronological age, but individuals are predicted to be somewhat older or younger than their known ages. Surprisingly, these deviations are not explained by the strongest predictors of lifespan in this population, early adversity and social integration. Instead, they are best predicted by male dominance rank: high-ranking males are predicted to be older than their true ages, and epigenetic age tracks changes in rank over time. Our results argue that achieving high rank for male baboons – the best predictor of reproductive success – imposes costs consistent with a 'live fast, die young' life-history strategy.

## Introduction

Aging, the nearly ubiquitous functional decline experienced by organisms over time (*López-Otín et al., 2013*), is a fundamental component of most animal life histories (*Jones et al., 2014*). At a physiological level, age affects individual quality, which in turn affects the ability to compete for mates and other resources, invest in reproduction, and maintain somatic integrity. At a demographic level, age is often one of the strongest predictors of survival and mortality risk, which are major determinants of Darwinian fitness. In order for patterns of aging to evolve, individuals must vary in their rates of biological aging. Thus, characterizing variation in biological aging rates and its sources – beyond simply chronological age – is an important goal in evolutionary ecology, with the potential to offer key insight into the trade-offs that shape individual life-history strategies (*Monaghan et al., 2008*).

Recent work suggests that DNA methylation data can provide exceptionally accurate estimates of chronological age (*Horvath and Raj, 2018*). These approaches typically use supervised machine

**eLife digest** For most animals, age is one of the strongest predictors of health and survival, but not all individuals age at the same rate. In fact, animals of the same species can have different 'biological ages' even when they have lived the same number of years. In humans and other mammals this variation in aging shows up in chemical modifications known as DNA methylation marks. Some researchers call these marks 'epigenetic', which literally means 'upon the genes'. And some DNA methylation marks change with age, so their combined pattern of change is often called the 'epigenetic clock'.

Environmental stressors, such as smoking or lack of physical activity, can make the epigenetic clock 'tick' faster, making the DNA of some individuals appear older than expected based on their actual age in years. These 'biologically older' individuals may also experience a higher risk of age-related disease. Studies in humans have revealed some of the reasons behind this fast biological aging, but it is unclear whether these results apply in the wild. It is possible that early life events trigger changes in the epigenetic clock, affecting health in adulthood. In primates, for example, adversity in early life has known effects on fertility and survival. Low social status also has a negative effect on health.

To find out whether early experiences and the social environment affect the epigenetic clock, Anderson, Johnston et al. tracked DNA methylation marks in baboons. This revealed that epigenetic clocks are strong predictors of age in wild primates, but neither early adversity nor the strength of social bonds affected the rate at which the clocks ticked. In fact, it was competition for social status that had the most dramatic effect on the clock's speed. Samples of males taken at different times during their lives showed that their epigenetic clocks sped up or slowed down as they moved up or down the social ladder, reflecting recent social experiences, rather than events early in their lives. On average, epigenetic clock measurements overestimated the age in years of alpha males by almost a year, showing that fighting to be on top comes at a cost.

This study highlights one way in which the social environment can influence aging. The next step is to understand how health is affected by the ways that animals attain social status. This could help researchers who study evolution understand how social interactions and environmental conditions affect survival and reproduction. It could also provide insight into the effects of social status on human health and aging.

learning methods that draw on methylation data from several hundred CpG sites, identified from hundreds of thousands of possible sites, to produce a single chronological age prediction (*Hannum et al., 2013*; *Horvath, 2013*; *Levine et al., 2018*). Intriguingly, some versions of these clocks also predict disease risk and mortality, suggesting that they capture aspects of biological aging that are not captured by chronological age alone (*Declerck and Vanden Berghe, 2018*). For example, in humans, individuals predicted to be older than their true chronological age are at higher risk of Alzheimer's disease (*Levine et al., 2015*), cognitive decline (*Levine et al., 2015*; *Marioni et al., 2015*), and obesity (*Horvath et al., 2014*). Accelerated epigenetic age is in turn predicted by environmental factors with known links to health and lifespan, including childhood social adversity (*Jovanovic et al., 2017*; *Raffington et al., 2020*) and cumulative lifetime stress (*Zannas et al., 2015*). These observations generalize to other animals. Dietary restriction, for instance, decelerates biological aging based on DNA methylation clocks developed for laboratory mice and captive rhesus macaques, and genetic knockout mice with extended lifespans also appear epigenetically young for age (*Maegawa et al., 2017*; *Petkovich et al., 2017*; *Stubbs et al., 2017*). However, while DNA methylation data have been used to estimate the age structure of wild populations (where birthdates are frequently unknown) (*De Paoli-Iseppi, 2018*; *Polanowski et al., 2014*; *Thompson et al., 2017*; *Wright et al., 2018*), they have not been applied to investigating sources of variance in biological aging in the wild.

To do so here, we coupled genome-wide data on DNA methylation levels in blood with detailed behavioral and life-history data available for one of the most intensively studied wild mammal populations in the world, the baboons of the Amboseli ecosystem of Kenya (*Alberts and Altmann, 2012*). First, we calibrated a DNA methylation-based 'epigenetic clock' and assessed the clock's

composition. Second, we compared the accuracy of this clock against other age-associated traits and between sexes. Third, we tested whether variance in biological aging was explained by socioenvironmental predictors known to impact fertility or survival in this population. Finally, we investigated an intriguing association between epigenetic age acceleration and male dominance rank. Our results show that predictors of lifespan can be decoupled from rates of epigenetic aging. However, other factors – particularly male dominance rank – are strong predictors of epigenetic clock-based age acceleration. These results are the first to establish a link between social factors and epigenetic aging in any natural animal population. Together, they highlight potential sex-specific trade-offs that may maximize fitness, but also compromise physiological condition and potentially shorten male lifespan.

## Results

### Epigenetic clock calibration and composition

We used a combination of previously published (*Lea et al., 2016*) and newly generated reduced-representation bisulfite sequencing (RRBS) data from 245 wild baboons (N = 277 blood samples) living in the Amboseli ecosystem of Kenya (*Alberts and Altmann, 2012*) to generate a DNA methylation-based age predictor (an 'epigenetic clock'; *Hannum et al., 2013*; *Horvath, 2013*). Starting with a data set of methylation levels for 458,504 CpG sites genome-wide (*Figure 1—figure supplement 1*; *Supplementary file 1*), we used elastic net regression to identify a set of 573 CpG sites that together accurately predict baboon age within a median absolute difference (MAD) of 1.1 years ± 1.9 s.d. (*Figure 1*; *Supplementary file 1*; Pearson's $r = 0.762$, p=$9.70 \times 10^{-54}$; median adult life expectancy in this population is 10.3 years for females and 7.9 for males; *Colchero et al., 2016*). The choice of these sites reflects a balance between increasing predictive accuracy within the sample and minimizing generalization error to unobserved samples, using a similar approach as that used to develop epigenetic clocks in humans (*Hannum et al., 2013*; *Horvath, 2013*) (see also Materials and methods and *Figure 1—figure supplement 2*).

Consistent with findings in humans (*Horvath, 2013*), clock sites are enriched in genes, CpG islands, promoter regions, and putative enhancers, compared to the background set of all sites we initially considered (i.e., the 458,504 CpG sites that were candidates for inclusion in the clock; in humans, this background set is the set of analyzable sites on the Illumina 27K methylation array [*Horvath, 2013*; *Figure 1—figure supplement 3*]; Fisher's exact tests, all p<0.05). Clock sites are also more common in age-associated differentially methylated regions in baboons (*Figure 1—figure supplement 3*; sites that increase with age: log2(odds ratio [OR])=4.189, p=$3.64 \times 10^{-9}$; sites that decrease with age: log2(OR)=5.344, p=$1.54 \times 10^{-8}$) (*Lea et al., 2015a*), such that many, but not all, of the clock sites also exhibit individual associations between DNA methylation levels and age (*Figure 1—figure supplement 4* and *Figure 2—figure supplement 1*; *Supplementary file 3*). Additionally, clock sites were more likely to be found in regions that exhibit enhancer-like activity in a massively parallel reporter assay (sites that increase with age: log2(OR)=2.685, p=$1.22 \times 10^{-2}$; sites that decrease with age: log2(OR)=4.789, p=$1.78 \times 10^{-5}$) (*Lea et al., 2018a*) and in regions implicated in the gene expression response to bacteria in the Amboseli baboon population (overlap of lipopolysaccharide [LPS] up-regulated genes and sites that increase with age: log2[OR]=0.907, p=$7.03 \times 10^{-4}$; overlap of LPS down-regulated genes and sites that decrease with age: log2[OR]=1.715, p=$1.55 \times 10^{-3}$) (*Lea et al., 2018b*). Our results thus suggest that the Amboseli baboon epigenetic clock not only tracks chronological age, but also captures age-related changes in blood DNA methylation levels that are functionally important for gene regulation, particularly in relation to the immune system.

### Comparison with other age-associated traits and differences between sexes

Overall, the clock performed favorably relative to other morphological or biomarker predictors of age in this population. The epigenetic clock generally explained more variance in true chronological age, resulted in lower median error, and exhibited less bias than predictions based on raw body mass index (BMI) or blood cell composition data from flow cytometry or blood smears (traits that change with age in baboons; *Altmann et al., 2010*; *Jayashankar et al., 2003*). Its performance was

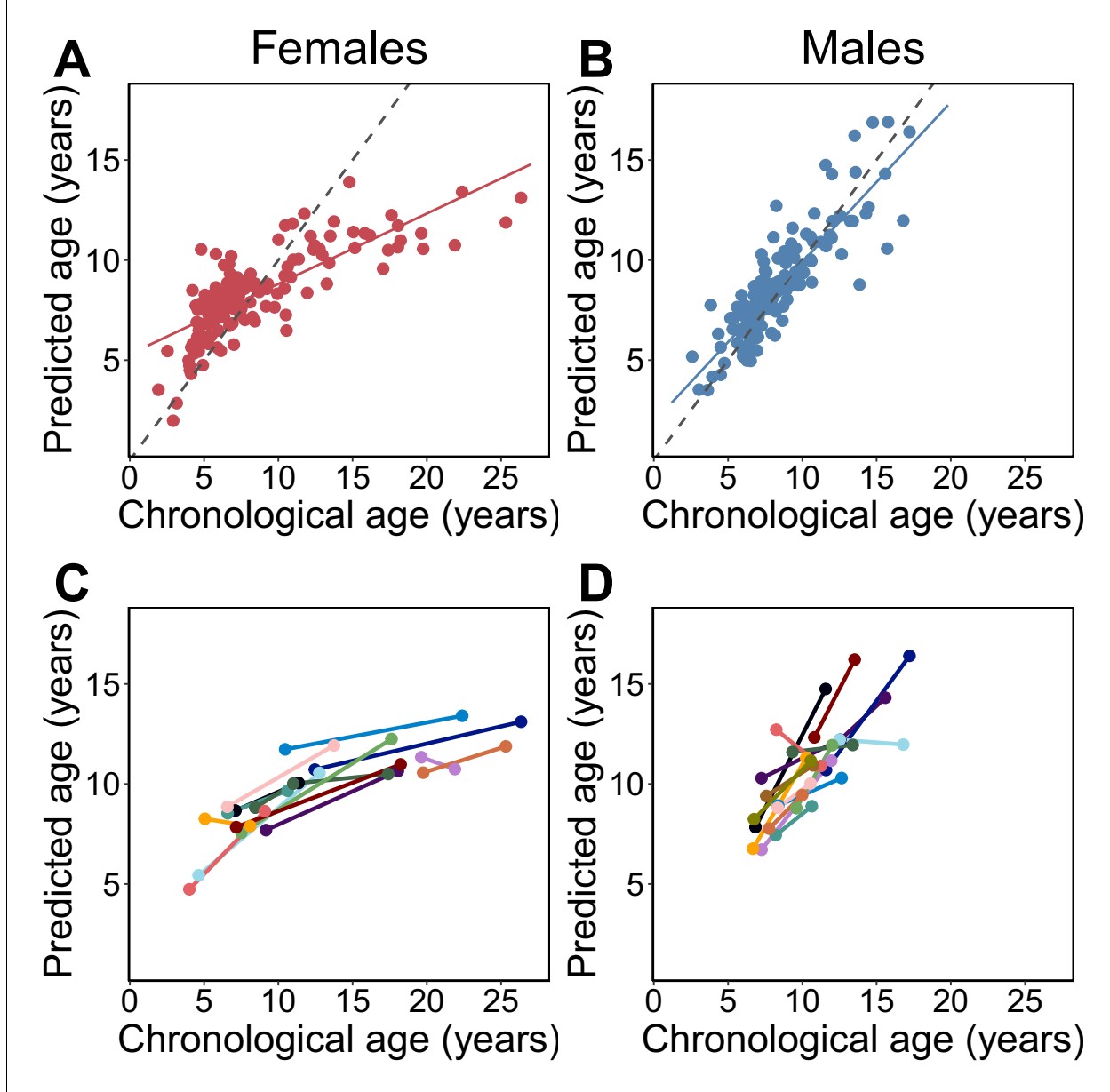

**Figure 1.** Epigenetic clock age predictions in the Amboseli baboons. Predicted ages are shown relative to true chronological ages for (A) females (Pearson's $r$ = 0.78, p=6.78×10⁻³⁰, N = 142 samples) and (B) males ($r$ = 0.86, p=5.49×10⁻⁴¹, N = 135 samples). Solid lines represent the best-fit line; dashed lines show the line for y = x. (C, D) Predictions for individuals with at least two samples in the data set (N = 30; 14 females and 16 males). In 26 of 30 cases (87%), samples collected later were correctly predicted to be from an older animal.

The online version of this article includes the following figure supplement(s) for figure 1:

**Figure supplement 1.** Characteristics of the RRBS data set.

**Figure supplement 2.** Comparison of clock performance across alternative values of alpha.

**Figure supplement 3.** Enrichment of the epigenetic clock CpG sites by genomic compartment.

**Figure supplement 4.** Association between age and DNA methylation level for individual clock CpG sites.

**Figure supplement 5.** Comparison of the performance of the epigenetic clock to other predictors of chronological age.

comparable to molar dentine exposure, a classical marker of age (*Galbany et al., 2011*; *Figure 1—figure supplement 5*). For a subset of 30 individuals, we had two samples collected at different points in time. The predicted ages from these longitudinally collected samples were older for the later-collected samples, as expected (*Figure 1C,D*; binomial test p=5.95×10⁻⁵). Furthermore, the

change in epigenetic clock predictions between successive longitudinal samples positively predicted the actual change in age between sample dates ($\beta$ = 0.312, p=0.027, controlling for sex; difference between actual change and predicted change: mean 3.11 years ± 3.25 s.d.).

However, clock performance was not equivalent in males and females. Specifically, we observed that the clock was significantly more accurate in males (*Figure 1*; males: N = 135; MAD = 0.85 years±1.0 s.d.; Pearson's $r$ = 0.86, p=5.49×10$^{-41}$; females: N = 142; MAD = 1.6 years±2.4 s.d.; $r$ = 0.78, p=6.78×10$^{-30}$; two-sided Wilcoxon test for differences in absolute error by sex: p=4.37×10$^{-9}$). Sex differences were also apparent in the slope of the relationship between predicted age and chronological age. Males show a 2.2-fold higher rate of change in predicted age, as a function of chronological age, compared to females (*Figure 1A,B*; chronological age by sex interaction in a linear model for predicted age: $\beta$ = 0.448, p=9.66×10$^{-19}$, N = 277). Interestingly, sex differences are not apparent in animals <8 years, which roughly corresponds to the age at which the majority of males have achieved adult dominance rank and dispersed from their natal group (*Alberts and Altmann, 1995a*; *Alberts and Altmann, 1995b*; *Alberts et al., 2003*) (N = 158, chronological age by sex interaction $\beta$ = −0.038, p=0.808). Rather, sex differences become apparent after baboons have reached full physiological and social adulthood (N = 119, chronological age by sex interaction $\beta$ = 0.459, p=9.74×10$^{-7}$ in animals ≥ 8 years), when divergence between male and female life-history strategies is most marked (*Alberts and Altmann, 1995a*; *Alberts and Altmann, 1995b*; *Alberts et al., 2003*) and when aging rates between the sexes are predicted to diverge (*Clutton-Brock and Isvaran, 2007*; *Kirkwood and Rose, 1991*; *Williams, 1957*).

Because of these differences, we separated males and females for all subsequent analyses. However, we note that the effects of age on DNA methylation levels at individual clock sites are highly correlated between the sexes (Pearson's $r$ = 0.91, p=3.35×10$^{-204}$), with effect sizes that are, on average, more precisely estimated in males (paired t-test p=4.53×10$^{-74}$ for standard errors of $\beta_{age}$; *Figure 1—figure supplement 4*). In other words, the sex differences in clock performance reflect changes in methylation that occur at the same CpG sites, but with higher variance in females. Lower accuracy in females compared to males therefore appears to result from the greater variability in DNA methylation change in older females (*Figure 1*).

## Socioenvironmental predictors of variance in biological aging

Although the baboon epigenetic clock is a good predictor of age overall, individuals were often predicted to be somewhat older or younger than their known chronological age. In humans and some model systems, the sign and magnitude of this deviation captures information about physiological decline and/or mortality risk beyond that contained in chronological age alone (*Maegawa et al., 2017*; *Petkovich et al., 2017*; *Stubbs et al., 2017*; *Ryan et al., 2020*).

To test whether this observation extends to wild baboons, we focused on four factors of known importance to fitness in the Amboseli population. First, we considered cumulative early adversity, which is a strong predictor of shortened lifespan and offspring survival for female baboons (*Tung et al., 2016*; *Zipple et al., 2019*). We measured cumulative adversity as a count of major adverse experiences suffered in early life, including low maternal social status, early-life drought, a competing younger sibling, maternal loss, and high experienced population density (i.e., social group size). Second, we considered social bond strength in adulthood, which positively predicts longer adult lifespan in baboons, humans, and other wild social mammals (*Archie et al., 2014a*; *Campos et al., 2020*; *Holt-Lunstad et al., 2010*; *Snyder-Mackler et al., 2020*). Third, we considered dominance rank, which is a major determinant of access to mates, social partners, and other resources in the Amboseli baboons (*Archie et al., 2014a*; *Alberts et al., 2006*; *Gesquiere et al., 2018*; *Lea et al., 2015b*). Finally, we considered BMI, a measure of body condition that, in the Amboseli baboons, primarily reflects lean muscle mass (mean body fat percentages have been estimated at <2% in adult females and <9% in adult males) (*Altmann et al., 1993*). Because raw BMI (i. e., BMI not correcting for age) also tracks growth and development (increasing as baboons reach their prime and then declining thereafter [*Altmann et al., 2010*; *Figure 2—figure supplement 2*]; Pearson's $r$ in males between rank and raw BMI = −0.56, p=6.38×10$^{-9}$), we calculated BMI relative to the expected value for each animal's age using piecewise regression, which also eliminates correlations between BMI and male rank (Pearson's $r$ = −0.070, p=0.504). We refer to this adjusted measure of BMI as age-adjusted BMI.

eLife Research article

Because high cumulative early adversity and low social bond strength are associated with increased mortality risk in the Amboseli baboons, we predicted that they would also be linked to increased epigenetic age. For rank and age-adjusted BMI, our predictions were less clear: improved resource access could conceivably slow biological aging, but increased investment in growth and reproduction (either through higher fertility in females or physical competition for rank in males) could also be energetically costly. To investigate these possibilities, we modeled the deviation between predicted age and known chronological age ($\Delta_{age}$) as a function of cumulative early adversity, ordinal dominance rank, age-adjusted BMI, and for females, social bond strength to other females. Social bond strength was not included in the model for males, as this measure was not available for a large proportion of males in this data set (53.8%). We also included chronological age as a predictor in the model, as epigenetic age tends to be systematically overpredicted for young individuals and underpredicted for old individuals (*Figure 1A,B*; this bias has been observed in both foundational work on epigenetic clocks [*Hannum et al., 2013*] and recent epigenetic clocks calibrated for rhesus macaques [*Horvath, 2020*], as well as for elastic net regression analyses more generally [*Engebretsen and Bohlin, 2019*]). Including chronological age in the model, as previous studies have done (*Hannum et al., 2013*; *Levine et al., 2018*), controls for this compression effect. None of the predictor variables were strongly linearly correlated (all Pearson's *r* < 0.35; *Supplementary file 4*).

Surprisingly, despite being two of the strongest known predictors of lifespan in wild baboons, neither cumulative early-life adversity nor social bond strength explain variation in $\Delta_{age}$ (*Table 1*). In contrast, high male dominance rank is strongly and significantly associated with larger values of $\Delta_{age}$ (β = −0.078, p=7.39×10$^{-4}$; *Figure 2*; *Table 1*; *Figure 2—figure supplement 3*). Alpha males are predicted to be an average of 10.95 months older than their true chronological age – a difference that translates to 11.5% of a male baboon's expected adult lifespan in Amboseli (*Colchero et al., 2016*). In contrast, dominance rank did not predict $\Delta_{age}$ in females (p=0.228; *Table 1*). Finally, age-adjusted BMI also predicted $\Delta_{age}$ in males (p=6.33×10$^{-3}$), but not in females (p=0.682; *Table 1*). These results are robust to inclusion of read depth and bisulfite conversion rate as covariates in the model (*Supplementary file 5*; in males, read depth is correlated with chronological age [$R^2$ = −0.409, p=0.038], but is not correlated with $\Delta_{age}$ [$R^2$ = 0.003, p=0.561]).

Despite the tendency for high-ranking males to have higher raw BMI due to increased muscle mass, the effects of rank and age-adjusted BMI in males are distinct. Specifically, modeling dominance rank after adjusting for raw BMI also produces a significant effect of rank on $\Delta_{age}$ in the same direction (p=9.93×10$^{-4}$; *Supplementary file 5*), as does substituting the age-adjusted BMI measure for either raw BMI or the residuals of raw BMI after adjusting for dominance rank (rank p=1.52×10$^{-2}$ and p=1.88×10$^{-4}$, respectively; *Supplementary file 5*). In contrast, BMI is only a significant predictor of male $\Delta_{age}$ when corrected for age (i.e., age-adjusted) and when rank is included in the same model (*Table 1*; *Supplementary file 5*). Furthermore, we obtain the same qualitative results if all low BMI males are removed from the sample (BMI < 41; this cut-off was used because it drops all young males who have clearly not reached full adult size; p=7.14×10$^{-3}$; *Supplementary file 5*). Dropping these males also eliminates the age-raw BMI correlation (Pearson's *r* = −0.16, p=0.21).

**Table 1.** Predictors of $\Delta_{age}$ *.

| Covariate | β (female) | p-value (female) | β (male) | p-value (male) |
|---|---|---|---|---|
| Intercept | 5.400 | 1.33 × 10$^{-15}$ | 3.294 | 1.19 × 10$^{-8}$ |
| Cumulative early adversity | −0.050 | 0.807 | −0.005 | 0.973 |
| Social bond strength | 0.382 | 0.164 | — | — |
| Dominance rank | 0.025 | 0.228 | −0.078 | 7.39 × 10$^{-4}$ |
| Age-adjusted BMI | 0.026 | 0.682 | 0.111 | 6.33 × 10$^{-3}$ |
| Chronological age | −0.699 | 1.62 × 10$^{-28}$ | −0.277 | 8.36 × 10$^{-8}$ |

*Separate linear models for $\Delta_{age}$ were fit for females (N = 66) and for males (N = 93) for whom no data values were missing; social bond strength was not included in the model for males. Significant results are shown in bold.

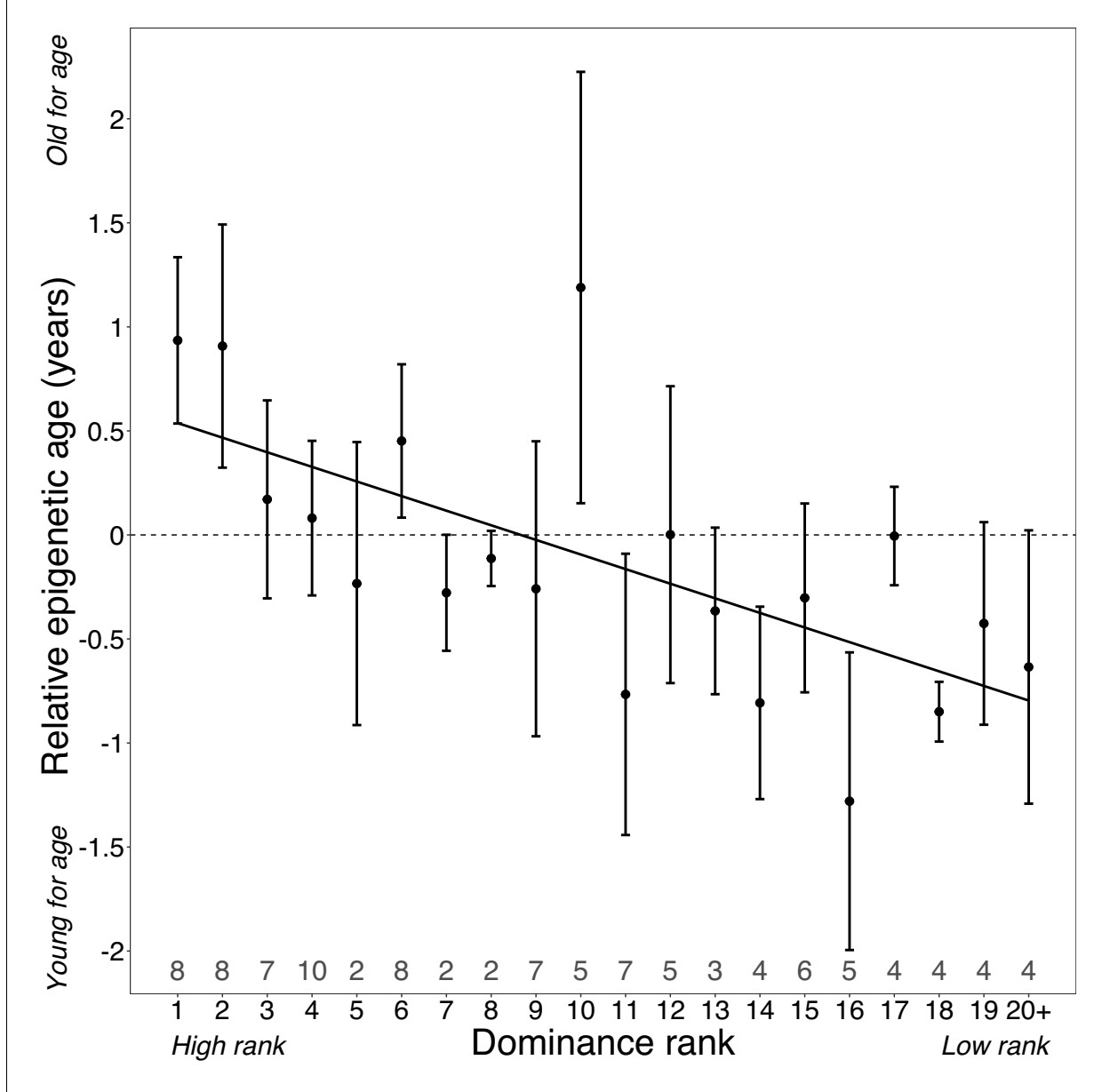

**Figure 2.** Dominance rank predicts relative epigenetic age in male baboons. High rank is associated with elevated values of $\Delta_{age}$ ($\beta$ = −0.0785, p=7.39×10$^{-4}$, N = 105). The y-axis shows relative epigenetic age, a measure of epigenetic aging similar to $\Delta_{age}$ that is based on the sample-specific residuals from the relationship between predicted age and true chronological age. Positive (negative) values correspond to predicted ages that are older (younger) than expected for that chronological age. Dominance rank is measured using ordinal values, such that smaller values indicate higher rank. Dots and error bars represent the means and standard errors, respectively. Gray values above the x-axis indicate sample sizes for each rank. The online version of this article includes the following figure supplement(s) for figure 2:

**Figure supplement 1.** Methylation levels of clock CpG sites across different genomic compartments.

**Figure supplement 2.** The relationship between age and body mass index in the Amboseli baboons.

**Figure supplement 3.** Relative epigenetic age versus chronological age.

**Figure supplement 4.** Male dominance rank versus chronological age.

## Male dominance rank predicts epigenetic age

In baboon males, achieving high rank depends on physical condition and fighting ability (*Alberts et al., 2003*). Consequently, rank in males is dynamic across the life course: males tend to attain their highest rank between 7 and 12 years of age and fall in rank thereafter (*Figure 2—figure*

supplement 4). Thus, nearly all males in the top four rank positions in our data set were between 7 and 12 years of age at the time they were sampled (however, because not all 7–12 year olds are high ranking, low-rank positions include males across the age range; *Supplementary file 1*, *Figure 2—figure supplement 4*). We therefore asked whether the association between high rank in males and accelerated epigenetic aging is a function of absolute rank values, regardless of age, or deviations from the *expected* mean rank given a male's age (i.e., 'rank-for-age'; *Figure 2—figure supplement 4*). We found that including rank-for-age as an additional covariate in the $\Delta_{age}$ model recapitulates the significant effect of ordinal male rank (p=0.045), but finds no effect of rank-for-age (p=0.819; *Supplementary file 5*). Our results therefore imply that males incur the costs of high rank primarily in early- to mid-adulthood, and only if they succeed in attaining high rank.

If attainment of high rank is linked to changes in epigenetic age within individuals, this pattern should be reflected in longitudinal samples. Specifically, males who improved in rank between samples should look older for age in their second sample relative to their first and vice versa. To assess this possibility, we calculated 'relative epigenetic age' (the residuals of the best-fit line relating chronological age and predicted age) for 14 males for whom we had repeated samples over time, 13 of whom changed ranks across sample dates (N = 28 samples, two per male). Samples collected when males were higher status predicted higher values of relative epigenetic age compared to samples collected when they were lower status (*Figure 3*; paired t-test, t = −2.99, p=0.011). For example, in the case of a male whom we first sampled at low status (ordinal rank = 18) and then after he had attained the alpha position (ordinal rank 1), the actual time that elapsed between samples was 0.79 years, but he exhibited an increase in *predicted* age of 2.6 years. Moreover, the two males that showed a decrease in predicted age, despite increasing in chronological age (*Figure 1D*), were among those that experienced the greatest drop in social status between samples. Thus, change in rank between samples for the same male predicts change in $\Delta_{age}$, controlling for chronological age ($R^2$ = 0.37, p=0.021). Consistent with our cross-sectional results, we found a suggestive relationship between change in $\Delta_{age}$ and BMI ($R^2$ = 0.31, p=0.08). Here, too, the effect of dominance rank does not seem to be driven by BMI: while the association between change in $\Delta_{age}$ and change in rank is no longer significant when modeling rank after adjusting for raw BMI, the correlation remains consistent ($R^2$ = 0.20, p=0.167). In contrast, raw BMI adjusted for rank explains almost none of the variance in change in $\Delta_{age}$ ($R^2$ = 0.01, p=0.779).

## Discussion

Together, our findings indicate that major environmental predictors of lifespan and mortality risk – particularly social bond strength and early-life adversity in this population – do not necessarily predict epigenetic measures of biological age. Although this assumption is widespread in the literature, including for epigenetic clock analyses (*Liu et al., 2019*; *Shalev and Belsky, 2016*), our results are broadly consistent with empirical results in humans. Specifically, while studies of early-life adversity, which also predicts lifespan in human populations, find relatively consistent support for a relationship between early adversity and accelerated epigenetic aging in children and adolescents (*Jovanovic et al., 2017*; *Raffington et al., 2020*; *Brody et al., 2016a*; *Brody et al., 2016b*; *Davis et al., 2017*; *Marini, 2018*; *Sumner et al., 2019*), there is little evidence for the long-term effects of early adversity on epigenetic age in adulthood (*Zannas et al., 2015*; *Austin et al., 2018*; *Boks et al., 2015*; *Lawn et al., 2018*; *Simons et al., 2016*; *Wolf et al., 2018*). Thus, while DNA methylation may make an important contribution to the biological embedding of early adversity into adulthood (*Aristizabal et al., 2020*; *Hertzman, 2012*), it does not seem to do so through affecting the epigenetic clock itself. Social and environmental effects on the clock instead seem to be most influenced by concurrent conditions, lending support to 'recency' models for environmental effects on aging that posit that health is more affected by the current environment than past experience (*Ben-Shlomo and Kuh, 2002*; *Shanahan et al., 2011*; *Shanahan and Hofer, 2011*). Additional longitudinal sampling will be necessary to evaluate whether current conditions alone can explain accelerated epigenetic aging or whether it also requires integrating the effects of exposures across the life course (the 'accumulation' model; *Ben-Shlomo and Kuh, 2002*; *Shanahan and Hofer, 2011*). Alternatively, the effects of early-life adversity and social bond strength may act through entirely distinct pathways than those captured by our epigenetic clock (including targeting tissues or cell types that we were unable to assess here). Indeed, the proliferation of alternative epigenetic clocks in humans

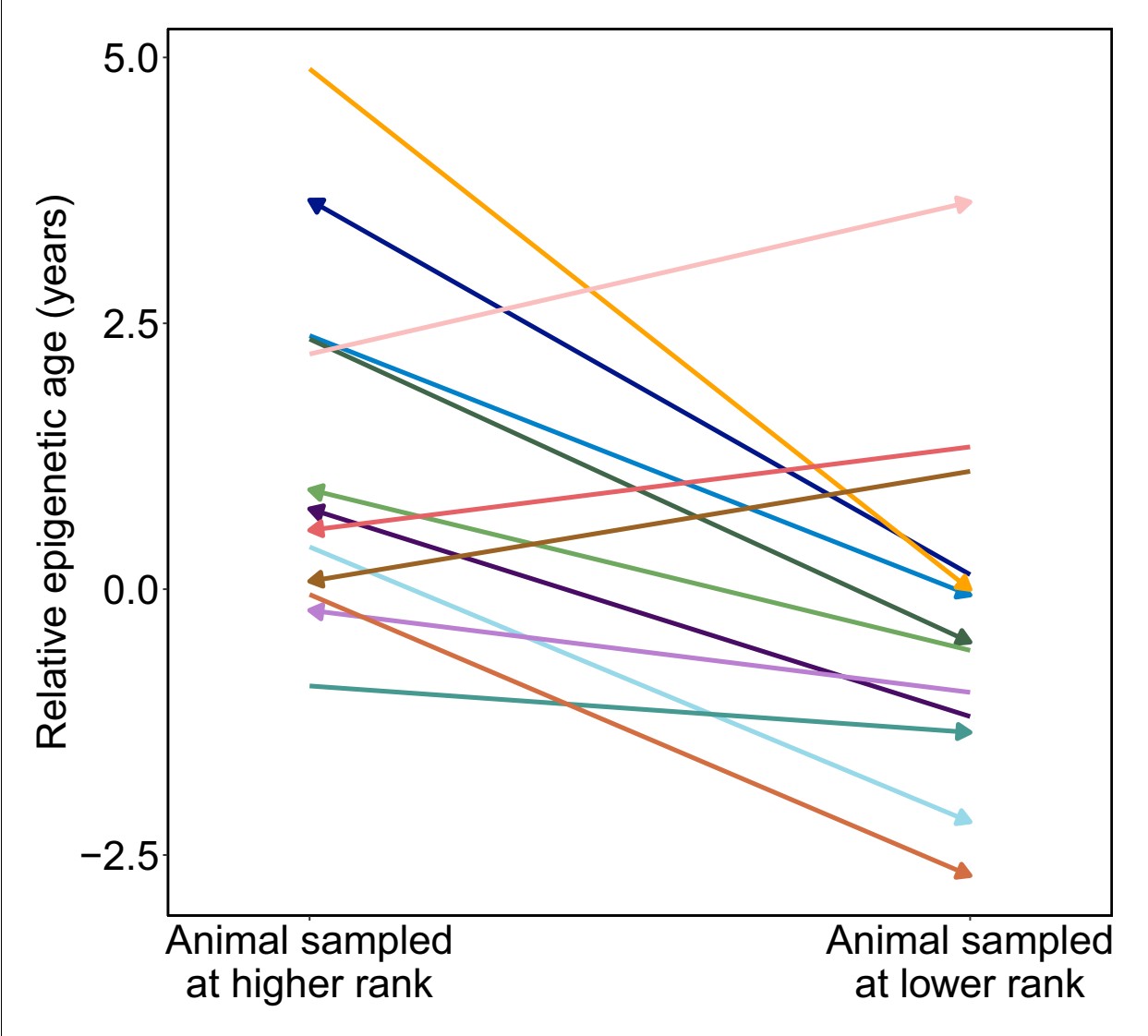

**Figure 3.** Male baboons exhibit higher relative epigenetic age when they occupy higher ranks. Relative epigenetic age for males in which multiple samples were collected when they occupied different ordinal rank values. Arrow indicates the temporal direction of rank changes: left-facing arrows represent cases in which the later sample was collected when males were higher ranking, and right-facing arrows represent cases in which the later sample was collected when males were lower ranking.

has revealed that the clocks that best predict chronological age are not necessarily the clocks that most closely track environmental exposures, and the same is likely to be true in other species (**Levine et al., 2018**; **Belsky et al., 2020**). Notably, the functional significance of the clock – that is, whether it reflects the mechanisms that causally drive aging, or instead serves as a passive biomarker – also remains unclear.

We found that the most robust socioenvironmental predictor of epigenetic age in the Amboseli baboons is male dominance rank, with a secondary effect of age-adjusted BMI observable when rank is included in the same model. Although high BMI also predicts accelerated epigenetic age in some human populations (**Ryan et al., 2020**), high BMI in these human populations is related to being overweight or obese. In contrast, because wild-feeding baboons in Amboseli are extremely lean (**Altmann et al., 1993**), the range of BMI in most human populations is distinct from the range exhibited by our study subjects (importantly, BMI in humans is calculated differently than BMI in baboons [see Materials and methods], and therefore the BMI scales are species specific). Instead, the higher BMI values in our dataset represent baboons in better body condition (more muscle

mass). Given that rank in male baboons is determined by physical fighting ability (*Alberts et al., 2003*), these results suggest that investment in body condition incurs physiological costs that accelerate biological age. If so, the rank effect we observe may be better interpreted as a marker of competitiveness, not as a consequence of being in a 'high rank' environment. In support of this idea, work on dominance rank and gene expression levels in the Amboseli baboons suggests that gene expression differences associated with male dominance rank tend to precede attainment of high rank, rather than being a consequence of behaviors exhibited after high rank is achieved (*Lea et al., 2018b*). Consistent with potential costs of attaining or maintaining high status, alpha males in Amboseli also exhibit elevated glucocorticoid levels (*Gesquiere et al., 2011*), increased expression of genes involved in innate immunity and inflammation (*Lea et al., 2018b*), and a trend toward elevated mortality risk (*Campos et al., 2020*). Males who can tolerate these costs and maintain high rank are nevertheless likely to enjoy higher lifetime reproductive success, since high rank is the single best predictor of mating and paternity success in baboon males (*Alberts et al., 2003*).

This interpretation may also explain major sex differences in the effects of rank on epigenetic age, where dominance rank shows no detectable effect in females. Dominance rank in female baboons is determined by nepotism, not physical competition: females typically insert into rank hierarchies directly below their mothers, and hierarchies therefore tend to remain stable over time (and even intergenerationally) (*Hausfater et al., 1982*). Our results contribute to an emerging picture in which dominance rank effects on both physiological and demographic outcomes are asymmetrical across sexes, and larger in males. Specifically, in addition to $\Delta_{age}$, male rank is a better predictor of immune cell gene expression and glucocorticoid levels than female rank (*Lea et al., 2018b*; *Gesquiere et al., 2011*; *Levy et al., 2020*). Recent findings suggest that high rank may also predict increased mortality risk in male Amboseli baboons, whereas neither high rank nor low rank predicts increased mortality risk in females (*Campos et al., 2020*). Together, these results argue that social status/dominance rank effects should not be interpreted as a universal phenomenon. Instead, the manner through which social status is achieved and maintained is likely to be key to understanding its consequences for physiology, health, and fitness (*Simons and Tung, 2019*). Specifically, we predict that high status will be most likely to accelerate the aging process, including epigenetic age, in species-sex combinations where high status increases reproductive success or fecundity, and achieving status is energetically costly (e.g., male red deer, mandrills, and geladas; female meerkats *Clutton-Brock et al., 2006*; *Clutton-Brock and Huchard, 2013*; *Emery Thompson and Georgiev, 2014*). Expanding studies of biological aging to a broader set of natural populations, especially those for which behavioral and demographic data are also available, will be key to testing these predictions.

## Materials and methods

### Study population and biological sample collection

This study focused on a longitudinally monitored population of wild baboons (*Papio cynocephalus*, the yellow baboon, with some admixture from the closely related anubis baboon *P. anubis Alberts and Altmann, 2001*; *Tung et al., 2008*) in the Amboseli ecosystem of Kenya. This population has been continuously monitored by the Amboseli Baboon Research Project (ABRP) since 1971 (*Alberts and Altmann, 2012*). For the majority of study subjects (N = 242 of 245 individuals), birth dates were therefore known to within a few days' error; for the remaining three individuals, birth dates were known within 3 months' error (*Supplementary file 1*).

All DNA methylation data were generated from blood-derived DNA obtained during periodic darting efforts, as detailed in *Lea et al., 2018b*; *Altmann et al., 1996*; *Tung et al., 2015*. Samples were obtained under approval from the Institutional Animal Care and Use Committee (IACUC) of Duke University (currently #A044-21-02) and adhered to all the laws and regulations of Kenya. In brief, individually recognized study subjects were temporarily anesthetized using a Telazol-loaded dart delivered through a blow gun. Baboons were then safely moved to a new location where blood samples and morphometric data, including body mass and crown-rump length, were collected. Baboons were then allowed to recover from anesthesia in a covered holding cage and released to their group within 2–4 hr. Blood samples were stored at −20° C in Kenya until export to the United States.

## DNA methylation data

DNA methylation data were generated from blood-extracted DNA collected from known individuals in the Amboseli study population (N = 277 samples from 245 animals; 13 females and 15 males were each sampled twice, and 1 female and 1 male were each sampled three times). Here, we analyzed a combined data set that included previously published RRBS (*Meissner et al., 2005*) data from the same population (N = 36 samples) (*Lea et al., 2016*) and new RRBS data from 241 additional samples.

RRBS libraries were constructed following *Boyle et al., 2012*, using ~200 ng baboon DNA plus 0.2 ng unmethylated lambda phage DNA per sample as input. Samples were sequenced to a mean depth of 17.8 (±10.5 s.d.) million reads on either the Illumina HiSeq 2000 or HiSeq 4000 platform (*Supplementary file 1*), with an estimated mean bisulfite conversion efficiency (based on the conversion rate of lambda phage DNA) of 99.8% (minimum = 98.1%). Sequence reads were trimmed with Trim Galore! (*Krueger, 2012*) to remove adapters and low quality sequence (Phred score < 20). Trimmed reads were mapped with BSMAP (*Xi and Li, 2009*) to the baboon genome (*Panu2.0*), allowing a 10% mismatch rate to account for the degenerate composition of bisulfite-converted DNA. We used autosomally mapped reads to count the number of methylated and total reads per CpG site, per sample (*Xi and Li, 2009*). To control for possible local genetic variation, we used BSMAP's rescaled 'effective total counts' measures, which adjusts for the presence of possible CpG site disrupting genetic variants. Importantly, although our population consists of hybrids, previous work on DNA methylation variation across baboon species shows that species differences have a negligible effect on quantifying DNA methylation (i.e., the rate of incorrect calls differs by <0.4% between anubis and yellow baboons, the two species that contribute to ancestry in Amboseli; *Vilgalys et al., 2019*).

Following *Lea et al., 2016*; *Lea et al., 2015a*, CpG sites were filtered to retain sites with a mean methylation level between 0.1 and 0.9 (i.e., to exclude constitutively hyper- or hypo-methylated sites) and mean coverage of $\geq 5\times$. We also excluded any CpG sites with missing data for $\geq 5\%$ of individuals in the sample. After filtering, we retained N = 458,504 CpG sites for downstream analysis. For the remaining missing data (mean number of missing sites per sample = 1.4 ± 3.5% s.d., equivalent to 6409 ± 16,024 s.d. sites), we imputed methylation levels using a k-nearest neighbors approach in the R package *impute*, using default parameters (*Hastie et al., 2001*).

## Building the epigenetic clock

We used the R package *glmnet* (*Friedman et al., 2009*) version 2.0.10 to build a DNA methylation clock for baboons. Specifically, we fit a linear model in which the predictor variables were normalized levels of DNA methylation at 458,504 candidate clock CpG sites across the genome and the response variable was chronological age. To account for the excess of CpG sites relative to samples, *glmnet* uses an elastic net penalty to shrink predictor coefficients toward 0 (*Friedman et al., 2010*). Optimal alpha parameters were identified by grid searching across a range of alphas from 0 (equivalent to ridge regression) to 1 (equivalent to Lasso) by increments of 0.1, which impacts the number of clock CpG sites by varying the degree of shrinkage of the predictor coefficients toward 0 (*Figure 1—figure supplement 2*). We defined the optimal alpha as the value that maximized $R^2$ between predicted and true chronological age across all samples. We set the regularization parameter lambda to the value that minimized mean-squared error during n-fold internal cross-validation.

To generate predicted age estimates for a given sample, we used a leave-one-out cross-validation approach in which all samples but the 'test' sample were included for model training, and the resulting model was used to predict age for the left-out test sample. To avoid leaking information from the training set into the test set, and to maximize the generalizability of the clock, we did not remove batch effects from the quantile normalized methylation ratio estimates. However, we confirmed that our results in the main model, for both males and for females, were robust if we added batch effect (previously generated samples [n = 36] versus newly generated samples [n = 241]) as a covariate. Training samples were scaled independently of the test sample in each leave-one-out model to avoid bleed-through of information from the test data into the training data. To do so, we first quantile normalized methylation ratios (the proportion of methylated counts to total counts for each CpG site) within each sample to a standard normal distribution. Training samples were then separated from the test sample and the methylation levels for each CpG site in the training set were

quantile normalized across samples to a standard normal distribution. To predict age in the test sample, we compared the methylation value for each site in the test sample to the empirical cumulative distribution function for the training samples (at the same site) to estimate the quantile in which the training sample methylation ratio fell. The training sample was then assigned the same quantile value from the standard normal distribution using the function *qnorm* in R.

## Epigenetic clock enrichment analyses

To evaluate whether CpG sites included in the epigenetic clock, relative to the 458,504 CpG background sites, were enriched in functionally important regions of the baboon genome (*Lea et al., 2015a*; *Vilgalys et al., 2019*), we used two-sided Fisher's exact tests to investigate enrichment/depletion of the 573 epigenetic clock sites in (1) gene bodies and exons, based on the Ensembl annotation *Panu2.0.90*; (2) CpG islands annotated in the UCSC Genome Browser; (3) CpG shores, defined as the 2000 basepairs flanking CpG islands (following *Lea et al., 2015a*; *Vilgalys et al., 2019*; *Irizarry et al., 2009*); and (4) promoter regions, defined as the 2000 basepairs upstream of the 5′-most annotated transcription start site for each gene (following *Lea et al., 2015a*; *Vilgalys et al., 2019*). We also considered (5) putative enhancer regions, which have not been annotated for the *Panu2.0* assembly. We therefore used ENCODE H3K4me1 ChIP-seq data from human peripheral blood mononuclear cells (PBMCs) (*ENCODE Project Consortium, 2012*) and the *liftOver* tool to define likely enhancer coordinates in *Panu2.0*.

We also tested for enrichment of clock sites in regions of the genome that have been identified by previous empirical studies to be of special interest. First, we considered regions that likely have regulatory activity in blood cells, defined as all 200 base-pair windows that showed evidence of enhancer activity in a recently performed massively parallel reporter assay (*Lea et al., 2018a*). We used *liftOver* to identify the inferred homologous *Panu2.0* coordinates for these windows, which were originally defined in the human genome. Second, we defined age-related differentially methylated regions in the Amboseli baboons based on genomic intervals found, in previous analyses, to contain at least three closely spaced age-associated CpG sites (inter-CpG distance ≤1 kb), as described in *Lea et al., 2015a*. Third, because inflammatory processes involved in innate immunity are strongly implicated in the aging process, we defined LPS up-regulated and LPS down-regulated genes as those genes that were significantly differentially expressed (1% false discovery rate) between unstimulated Amboseli baboon white blood cells and LPS-stimulated cells from the same individual, following 10 hr of culture in parallel (*Lea et al., 2018b*).

## Comparisons to alternative predictors of aging

To assess the utility of the DNA methylation clock relative to other data types, we compared its predictive accuracy to clocks based on three other age-related phenotypes: tooth wear (percent molar dentine exposure; *Galbany et al., 2011*), body condition (BMI; *Altmann et al., 2010*), and blood cell type composition (blood smear counts and lymphocyte/monocyte proportions from flow cytometry performed on peripheral blood mononuclear cells, as in *Lea et al., 2018b*; *Snyder-Mackler et al., 2016*). Leave-one-out model training and prediction were performed for each data type using linear modeling. To compare the relative predictive accuracy of each data type, we calculated the $R^2$ between predicted and chronological age, the MAD between predicted and chronological age, and the bias in age predictions (the absolute value of 1 − slope of the best-fit line between predicted and chronological age) (*Figure 1—figure supplement 5*).

### Tooth wear

Molar enamel in baboons wears away with age to expose the underlying dentine layer. Percent dentine exposure (PDE) on the molar occlusal surface has been shown to be strongly age-correlated in previous work (*Galbany et al., 2011*). To assess its predictive power, we obtained PDE data from tooth casts reported by *Galbany et al., 2011* for the left upper molars (tooth positions M1, M2, M3) and left lower molars (tooth positions M1, M2, M3) for 39 males and 34 females in our data set. For each molar position (M1, M2, M3) within each individual, we calculated PDE as the mean for the upper and lower molars. Because dentine exposure scales quadratically with respect to age (*Galbany et al., 2011*), we fit age as a function of PDE using the following model:

$$age \sim \sqrt{PDE_{M1}} + \sqrt{PDE_{M2}} + \sqrt{PDE_{M3}}$$

### Body mass index

For both male and female baboons in Amboseli, body mass increases with age until individuals reach peak size and then tends to decrease with age as animals lose fat and/or muscle mass (*Altmann et al., 2010*). To quantify body condition using body mass, we calculated BMI values for 139 males and 154 females for whom body mass and crown-rump length data were available from periodic darting efforts. We retained only measures taken from animals born into and sampled in wild-feeding study groups, when sex-skin swellings (in females only) that could affect crown-rump length measures were absent. BMI was calculated as mass (kilograms) divided by crown-rump length (meters squared), following *Altmann et al., 1993*. To assess the predictive power of age-adjusted BMI, we built sex-specific piecewise-regression models using the package *segmented* in R (*Muggeo and Muggeo, 2017*). Breakpoints for the piecewise-regression models (to separate 'youthful' versus 'aged' animals) were initialized at 8 years old for males and 10 years old for females, following findings from previous work on body mass in the Amboseli population (*Altmann et al., 2010*).

### Blood cell type composition

The proportions of different cell types in blood change across the life course, including in baboons (*Jayashankar et al., 2003*). We assessed the predictive power of blood cell composition for age using two data sets. First, we used data collected from blood smear counts (N = 134) for five major white blood cell types: basophils, eosinophils, monocytes, lymphocytes, and neutrophils. Second, we used data on the proportional representation of five PBMC subsets: cytotoxic T cells, helper T cells, B cells, monocytes, and natural killer cells, measured using flow cytometry as reported by *Lea et al., 2018b* (N = 53). Cell types were included as individual covariates for leave-one-out model training.

## Sources of variance in predicted age

We asked whether factors known to be associated with inter-individual variation in fertility or survival also predict inter-individual variation in $\Delta_{age}$ (predicted age from the epigenetic clock minus known chronological age). To do so, we fit linear models separately for males and females, with $\Delta_{age}$ as the dependent variable and dominance rank at the time of sampling, cumulative early adversity, age-adjusted BMI, and chronological age as predictor variables (*Tung et al., 2016*). For females, we also included a measure of social bond strength to other females as a predictor variable, based on findings that show that socially isolated females experience higher mortality rates in adulthood (*Archie et al., 2014a*; *Silk et al., 2010*). Samples with missing values for any of the predictor variables were excluded in the model, resulting in a final analysis set of 66 female samples (from 59 females) and 93 male samples (from 84 males). The chronological ages of samples with complete data relative to samples with missing data were equivalent for females (t-test, t = 1.95, p=0.053) but were slightly lower for males (t-test, t = −3.04, p=0.003; mean chronological ages are 7.98 and 9.65 years for complete and missing samples, respectively). Predictor variables were measured as follows.

### Dominance rank

Sex-specific dominance hierarchies were constructed monthly for every social group in the study population based on the outcomes of dyadic agonistic encounters. An animal was considered to win a dyadic agonistic encounter if it gave aggressive or neutral, but not submissive, gestures, and the other animal gave submissive gestures only (*Hausfater, 1975*). These wins and losses were entered into a sex-specific data matrix, such that animals were ordered to minimize the number of entries falling below the matrix diagonal (which would indicate that the lower ranked individual won a dyadic encounter). Ordinal dominance ranks were assigned on a monthly basis to every adult based on these matrices, such that low numbers represent high rank/social status and high numbers represent low rank/social status (*Alberts et al., 2003*; *Hausfater et al., 1982*). Although most analyses of data from the Amboseli baboons have used ordinal ranks as the primary measure of social status, in some analyses proportional rank (i.e., the proportion of same-sex members of an individual's social group that he or she dominates) has proven to be a stronger predictor of other trait outcomes (*Archie et al., 2014b*; *Levy, 2020*). In this study, we chose to use ordinal ranks, but proportional

and ordinal dominance rank were highly correlated in this particular dataset ($R^2 = 0.70$, $p=1.13 \times 10^{-58}$). Using ordinal rank rather than proportional rank therefore did not qualitatively affect our analyses. Additionally, to investigate whether the patterns we observed are due to a consistent effect of rank across all ages, or instead an effect of being high or low rank relative to the expected (mean) value for a male's age, we also calculated a 'rank-for-age' value. Rank-for-age is defined as the residuals of a model with dominance rank as the response variable and age and age$^2$ as the predictor variables (*Figure 2—figure supplement 4*).

## Cumulative early adversity

Previous work in Amboseli defined a cumulative early adversity score as the sum of six different adverse conditions that a baboon could experience during early life (*Tung et al., 2016*). This index strongly predicts adult lifespan in female baboons, and a modified version of this index also predicts offspring survival (*Zipple et al., 2019*). To maximize the sample size available for analysis, we excluded maternal social connectedness, the source of adversity with the highest frequency of missing data, leaving us with a cumulative early adversity score generated from five different binary-coded adverse experiences. These experiences were as follows: (1) early-life drought (defined as ≤200 mm of rainfall in the first year of life), which is linked to reduced fertility in females (*Lea et al., 2015b*; *Beehner et al., 2006*); (2) having a low ranking mother (defined as falling within the lowest quartile of ranks for individuals in the data set), which predicts age at maturation (*Altmann and Alberts, 2003a*; *Altmann et al., 1988*; *Charpentier et al., 2008*); (3) having a close-in-age younger sibling (<1.5 years), which may redirect maternal investment to the sibling (*Altmann et al., 1978*), (4) being born into a large social group, which may increase within-group competition for shared resources (*Lea et al., 2015b*; *Charpentier et al., 2008*; *Altmann and Alberts, 2003b*), and (5) maternal death before the age of 4, which results in a loss of both social and nutritional resources (*Charpentier et al., 2008*; *Lea et al., 2014*).

## Body mass index

Age-adjusted BMI was modeled as the residuals from sex-specific piecewise-regression models relating raw BMI to age. By taking this approach, we asked whether having relatively high BMI for one's age and sex predicted higher (or lower) $\Delta_{age}$. To calculate rank-adjusted BMI values, we modeled raw BMI as a function of rank in a linear model and calculated the residuals from the model. To calculate dominance rank adjusted for raw BMI, we took the inverse approach. We note that BMI for baboons is not directly comparable to BMI for humans because baboon BMI is measured as body mass divided by the square of crown-rump length (because baboons are quadrupedal), whereas human BMI is calculated as body mass divided by the square of standing height.

## Social bond strength

For this analysis, we measured female social bond strength to other females using the dyadic sociality index (DSI$_F$) (*Campos et al., 2020*). We did not include this parameter (male's social bond strength to females) for the male model because this measure is unavailable for many males in this data set. DSI$_F$ was calculated as an individual's average bond strength with her top three female social partners, in the 365 days prior to the day of sampling, controlling for observer effort. This approach is based on representative interaction sampling of grooming interactions between females, in which observers record all grooming interactions in their line of sight while moving through the group conducting random-ordered, 10 min long focal animal samples of pre-selected individuals. Because smaller groups receive more observer effort per individual and per dyad (and thus record more grooming interactions per individual or dyad), we estimated observer effort for dyad $d$ in year $y$ as:

$$E_{d,y} = \frac{c_d(s_d)}{f_d}$$

where $c_d$ is the number of days the two females in a dyad were coresident in the same social group, $s_d$ is the number of focal samples taken during the dyad's coresidence, and $f_d$ is the average number of females in the group during the dyad's coresidence.

DSI$_F$ for each adult female dyad in each year is the z-scored residual, $\varepsilon$, from the model:

$$log(R_{d,y}) = \beta\left(\log\left(E_{d,y}\right)\right) + \varepsilon$$

where $R_{d,y}$ is the number of grooming interactions for dyad $d$ in year $y$ divided by the number of days that the two individuals were coresident, and $E_{d,y}$ is observer effort.

## Analysis of longitudinal samples

To test whether changes in rank predict changes in relative epigenetic age within individuals, we used data from 11 males from the original data set and generated additional RRBS data for nine samples, resulting in a final set of 14 males who each were sampled at least twice in the data set, 13 of whom occupied different ordinal ranks at different sampling dates (mean years elapsed between samples = 3.7 ± 1.65 s.d.; mean absolute difference in dominance ranks = 1.29 ± 8.34 s.d.). This effort increased our total sample size to N = 286 samples from 248 unique individuals. To incorporate the new samples into our analysis, we reperformed leave-one-out age prediction with N-fold internal cross-validation at the optimal alpha selected for the original N = 277 samples (alpha = 0.1). For the 277 samples carried over from the original analysis, we verified that age predictions were nearly identical between the previous analysis and the expanded data set ($R^2$ = 0.98, p=$2.21 \times 10^{-239}$; *Supplementary file 1*).

Based on the new age predictions for males in the data set (N = 140), we again calculated relative epigenetic age as the residual of the best-fit line relating predicted age to chronological age. We then used the 14 males with repeated DNA methylation profiles and rank measures in this data set to test whether, within individuals, changes in dominance rank or rank-for-age explained changes in relative epigenetic age between samples. In total, five males were sampled three times. For four of these five, we only included the two samples that were sampled the farthest apart in time (i.e., excluded the temporal middle sample) to maximize the age change between sample dates. For the fifth male, BMI information was missing for the third sample, so we included the first two samples collected in time.

## Code availability

All R code used to analyze data in this study is available at https://github.com/janderson94/BaboonEpigeneticAging; *Anderson, 2021*; with a copy archived at swh:1:rev: 58ca836d3416c2a447cbd055aee66c11140aec86.

## Acknowledgements

We gratefully acknowledge the support provided by the National Science Foundation and the National Institutes of Health for the majority of the data represented here, currently through NSF IOS 1456832, NIH R01AG053308, R01AG053330, R01HD088558, and P01AG031719. RAJ is supported by NIH F32HD095616 and JAA by NSF #2018264636. We also acknowledge support from the Canadian Institute of Advanced Research (Child and Brain Development Program); support for high-performance computing resources from the North Carolina Biotechnology Center (Grant Number 2016-IDG-1013); and a seed grant from the Center for Population Health and Aging (P30AG034424 to A O'Rand). We thank the members of the Amboseli Baboon Research Project for collecting the data presented here, especially J Altmann for her foundational role in establishing the study population and these data sets; J Gordon, N Learn, and K Pinc for managing the database; RS Mututua, S Sayialel, and JK Warutere for data collection in the field; and T Wango and V Oudu for their assistance in Nairobi. We also thank the Kenya Wildlife Service, University of Nairobi, the Institute of Primate Research, the National Museums of Kenya, the National Council for Science, Technology, and Innovation, members of the Amboseli-Longido pastoralist communities, the Enduimet Wildlife Management Area, Ker and Downey Safaris, Air Kenya, and Safarilink for their assistance in Kenya. Finally, we thank J Galbany for assistance with the molar dentine data set; current and past members of the Tung, Alberts, Archie, and Altmann labs for their helpful feedback; and J Higham, C Kuzawa, and three anonymous reviewers for constructive critiques of a previous version of this manuscript. This research was approved by IACUCs at Duke University, University of Notre Dame, and Princeton University and adhered to all the laws and regulations of Kenya. For a complete set of

acknowledgments of funding sources, logistical assistance, and data collection and management, please visit http://amboselibaboons.nd.edu/acknowledgements/.

# Additional information

## Competing interests

Jenny Tung: Reviewing editor, *eLife*. The other authors declare that no competing interests exist.

## Funding

| Funder | Grant reference number | Author |
| --- | --- | --- |
| National Science Foundation | IOS 1456832 | Susan C Alberts |
| National Institutes of Health | R01AG053308 | Susan C Alberts |
| National Institutes of Health | R01AG053330 | Elizabeth A Archie |
| National Institutes of Health | R01HD088558 | Jenny Tung |
| National Institutes of Health | P01AG031719 | Susan C Alberts |
| National Institutes of Health | F32HD095616 | Rachel A Johnston |
| National Science Foundation | 2018264636 | Jordan A Anderson |
| Foerster-Bernstein Foundation | Postdoctoral Fellowship | Rachel A Johnston |
| North Carolina Biotechnology Center | 2016-IDG-1013 | Jenny Tung |
| Center for Population Health and Aging | P30AG034424 | Jenny Tung |
| Canadian Institute for Advanced Research | | Jenny Tung |

The funders had no role in study design, data collection and interpretation, or the decision to submit the work for publication.

## Author contributions

Jordan A Anderson, Rachel A Johnston, Conceptualization, Formal analysis, Investigation, Writing - original draft, Writing - review and editing; Amanda J Lea, Conceptualization, Investigation, Writing - review and editing; Fernando A Campos, Tawni N Voyles, Mercy Y Akinyi, Investigation, Writing - review and editing; Susan C Alberts, Funding acquisition, Investigation, Writing - review and editing; Elizabeth A Archie, Conceptualization, Funding acquisition, Investigation, Writing - review and editing; Jenny Tung, Conceptualization, Supervision, Funding acquisition, Investigation, Writing - original draft, Writing - review and editing

## Author ORCIDs

Jordan A Anderson https://orcid.org/0000-0002-8109-7136
Rachel A Johnston https://orcid.org/0000-0002-8965-1162
Amanda J Lea http://orcid.org/0000-0002-8827-2750
Fernando A Campos http://orcid.org/0000-0001-9826-751X
Mercy Y Akinyi https://orcid.org/0000-0002-3835-5793
Susan C Alberts http://orcid.org/0000-0002-1313-488X
Elizabeth A Archie http://orcid.org/0000-0002-1187-0998
Jenny Tung https://orcid.org/0000-0003-0416-2958

## Ethics

Animal experimentation: Samples were obtained under approval from the Institutional Animal Care and Use Committee (IACUC) of Duke University (#A273-17-12) and adhered to all the laws and regulations of Kenya.

Decision letter and Author response
Decision letter https://doi.org/10.7554/eLife.66128.sa1
Author response https://doi.org/10.7554/eLife.66128.sa2

## Additional files

### Supplementary files

• Supplementary file 1. Samples used for generating the RRBS dataset.

• Supplementary file 2. Genomic coordinates, average annual methylation level change (averaged across the 277 leave-one-out models), and genomic context for the 573 CpG sites in the epigenetic clock.

• Supplementary file 3. Results from site-by-site models (for each of 534 testable clock sites) predicting chronological age, controlling for relatedness.

• Supplementary file 4. Pearson correlations among covariates for females (above diagonal) and males (below diagonal), with p-values in parentheses.

• Supplementary file 5. Linear models for Δage, fit for males, with significant results shown in bold.

• Transparent reporting form

### Data availability

All sequencing data generated during this study are available in the NCBI Sequence Read Archive (project accession PRJNA648767), and processed counts data are available in the Dryad digital repository (https://doi.org/10.5061/dryad.qjq2bvqf0).

The following datasets were generated:

| Author(s) | Year | Dataset title | Dataset URL | Database and Identifier |
|---|---|---|---|---|
| Anderson JA, Johnston RA, Lea AJ, Campos FA, Voyles TN, Akinyi MY, Alberts SC, Archie EA, Tung J | 2021 | Baboon Epigenetic Aging | https://www.ncbi.nlm.nih.gov/bioproject/PRJNA648767 | NCBI BioProject, PRJNA648767 |
| Anderson JA, Johnston RA, Lea AJ, Campos FA, Voyles TN, Akinyi MY, Alberts SC, Archie EA, Tung J | 2021 | High social status males experience accelerated epigenetic aging in wild baboons | https://doi.org/10.5061/dryad.qjq2bvqf0 | Dryad Digital Repository, 10.5061/dryad.qjq2bvqf0 |

The following previously published dataset was used:

| Author(s) | Year | Dataset title | Dataset URL | Database and Identifier |
|---|---|---|---|---|
| Lea AJ, Altmann J, Alberts SC, Tung J | 2015 | Papio cynocephalus Epigenomics | https://trace.ncbi.nlm.nih.gov/Traces/sra/?study=SRP058411 | NCBI BioProject, SRP058411 |

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
