## [Decision Letter]

**Acceptance summary:**

In this paper, the authors collect epigenomic data from a well-studied wild baboon community, which they use to construct an epigenetic clock, a method of measuring "biological age" that is increasingly used as a tool in human aging research. The authors find that deviations between biological and chronological age can in part be explained by social phenomena. In particular, for male baboons, maintaining social dominance may play an important role in accelerating the dimension of aging indexed by this measure. This is a foundational study for social-biological-health research.

**Decision letter after peer review:**

Thank you for submitting your article "High social status males experience accelerated epigenetic aging in wild baboons" for consideration by *eLife*. Your article has been reviewed by 2 peer reviewers, and the evaluation has been overseen by a Reviewing Editor and George Perry as the Senior Editor. The following individuals involved in review of your submission have agreed to reveal their identity: James Higham (Reviewer #1); Chris Kuzawa (Reviewer #2).

Essential Revisions:

1. Address comments from reviewer #4 from the previous journal.

2. Revise in response to the two minor comments/suggestions from Reviewer #2.

3. Consider whether residuals should be analyzed in the models or if it is better to set up the model differently, from a statistical best practices standpoint.

Reviewer #1 (Recommendations for the authors):

It has already been through multiple rounds of review at another journal, which have improved it substantively. While there are some outstanding comments to address from another reviewer, in my view the manuscript does not need much further work before it could be published in *eLife*.

Reviewer #2 (Recommendations for the authors):

I have been asked to review this manuscript, which has already gone through 3 rounds of review at another journal. I have had access to that set of reviews and the authors' responses, and read them, and the new manuscript draft, carefully. Of the three reviewers, two provided suggestions and were satisfied with the authors' revisions in addressing their points and recommended publication. One reviewer seems to have held up publication in that journal. That reviewer raised three main issues: around the possible confounding role of BMI, about whether to adjust for chronological age, and about the need to provide evidence of replication. As I will discuss in my brief comments here, I feel that the authors addressed all of these points effectively and thoroughly, and I feel that the paper should have been accepted for publication at the prior journal. I support publication at *eLife*, assuming that new methods/statistical points noted in a new review (by reviewer 4) can be addressed, once the authors have seen them and had a chance to respond to them.

Prior point #1: Does the BMI confound associations between rank and epigenetic age acceleration in males? Based upon findings in human studies, reviewer 1 argues that the higher BMIs of high ranking individuals could be the true cause of their apparent acceleration in biological age, rather than this being due to rank per se. The authors argue compellingly that the BMI in humans may tell us about adiposity (which can lead to metabolic derangement, inflammation etc – thereby potentially accelerating biological age measures), whereas in wild baboons (especially male) it is more informative about lean mass. I think the authors very thoroughly and effectively addressed these points, and I had the same reaction to the reviewer's critique when I read it.

Prior point #2: Should models control for chronological age? I found this critique hard to understand. In analyses of epigenetic clocks, it is acceleration – defined as deviations from expected, chronological age – that are generally of interest. This requires adjustment for chronological age by definition, so that what is being predicted are the residuals. In addition, some of the measures (BMI) change with age, and thus, not adjusting for age would convert those measures into age proxies. In my view, the authors carefully and thoroughly addressed the reviewer's concern in their responses to prior rounds of review.

Prior point #3: Validation in another species or population is necessary. The reviewer argued that papers published in high impact journals require replication. While that is the norm in certain fields (e.g. GWAS studies of the predictors of polygenic traits in humans), that standard is less applicable in the context of a study of a wild primate population. As the authors note, many (most, or almost all) high impact analyses of longitudinal studies of this sort do not include replication. In some instances, populations (like Amboslei) provide unique opportunities given their unusual design (in the case of Amboseli, more than half century duration). The analyses here require empirically measured ages tracked longitudinally. The idea of replicating in another species, as the reviewer suggests, misses the point made in a compelling way by the authors – the nature of hierarchy, related stressors etc. vary across species and demanding that similar relationships should replicate across species is to miss a big part of the story.

In sum, I felt that the authors effectively addressed the main critiques of the first round of review and that publication would have been warranted at that stage. They provided even more thorough and compelling responses in rounds 2 and 3. I am aware of a new set of comments, primarily from a statistical/methods angle, which the authors have not yet seen or addressed, and that are outside my area of expertise. Assuming that they can address those suggestions in a satisfactory way, I support publication of this analysis.

I think the manuscript in its current form is tight and compelling and is basically ready for publication. Because I anticipate that the authors may be revising in response to the new statistical review, I provide a few minor suggestions on how they might tighten and hone several of the points made in the manuscript:

I would suggest rereading the manuscript to make sure that it is clear for the reader that the clock they are reporting is measured specifically in blood – that is, immune cells, and should be interpreted in that light. Some past work has generated clocks that work across multiple tissue types, but that is not known here. I do not see this as at all problematic, and this is obviously the only ethical option when studying wild primates. But I do think the authors could be clearer about this when describing their findings and their functional significance: These findings speak specifically to markers of aging in immunity, and it is not clear whether they are informative about changes in other tissues that may not be harvested in wild primate populations.

Another, related point is the fact that the functional significance of epigenetic clocks (why they "work" at all) remains uncertain. The extent to which they are in the causal pathway to health outcomes (as suggested by at least one recent study that likely postdates this draft: Lu, Yuancheng, et al. "Reprogramming to recover youthful epigenetic information and restore vision." Nature 588.7836 (2020): 124-129), or merely a marker, is unclear. In the end, these are useful measures if they predict relevant outcomes, but I think noting this current ambiguity around their functional basis might warrant a sentence or two in the discussion.

---

## [Author Response]

Essential Revisions:1. Address comments from reviewer #4 from the previous journal.

We have addressed all comments from Reviewer #4 (see below for details).

2. Revise in response to the two minor comments/suggestions from Reviewer #2 (see below).

We have made revisions in response to both of Reviewer #2’s suggestions. Specifically, we now clarify that our analyses pertain specifically to DNA methylation in the peripheral blood (lines 64; 109-110) and highlight the importance of resolving the mechanistic implications of the epigenetic clock (if any) in future work (lines 302-304).

3. Consider whether residuals should be analyzed in the models or if it is better to set up the model differently, from a statistical best practices standpoint.

We incorporated residuals in our models in three places, which we have chosen to retain in the revised manuscript for the following reasons. First, for the purposes of our longitudinal analysis of male rank and epigenetic age, we defined “relative epigenetic age” as the residuals of the best fit line relating chronological age and predicted age (lines 257-259). We followed this approach because we were interested in testing whether samples looked “old-for-age” in males who improved their rank between samples (and vice-versa for males who fell in rank). In other words, our question was specifically about the sign and magnitude of residual error in the epigenetic age prediction. Importantly, the results of this analysis are concordant with our cross-sectional analysis of ∆_age_, which did not use residuals.

In the second instance, we defined a measure of “rank-for-age”—i.e., whether males were higher or lower rank than expected, given the typical age-related pattern for male rank—based on the residuals of rank after regressing out age (lines 243-246). In this case, we were again testing an explicit hypothesis about the sign and magnitude of residual error.

Finally, we considered the possibility that body mass index might drive the relationship between rank and epigenetic age. We found this explanation unlikely, but as it was a major concern of a previous reviewer, we evaluated whether our main model of ∆_age_ was perturbed by (i) replacing rank with the residuals of rank after regressing out raw BMI; (ii) replacing age-adjusted BMI with raw BMI; or (iii) replacing age-adjusted BMI with the residuals of raw BMI controlling for rank (lines 273-277). None of these alternative predictors changed our main findings, suggesting that the choice to model residuals does not actually matter to our results. However, the inclusion of these analyses may clarify the nature of the rank effect to readers who have the same concern as the reviewer.

Reviewer #2 (Recommendations for the authors):[…]I think the manuscript in its current form is tight and compelling and is basically ready for publication. Because I anticipate that the authors may be revising in response to the new statistical review, I provide a few minor suggestions on how they might tighten and hone several of the points made in the manuscript:I would suggest rereading the manuscript to make sure that it is clear for the reader that the clock they are reporting is measured specifically in blood – that is, immune cells, and should be interpreted in that light. Some past work has generated clocks that work across multiple tissue types, but that is not known here. I do not see this as at all problematic, and this is obviously the only ethical option when studying wild primates. But I do think the authors could be clearer about this when describing their findings and their functional significance: These findings speak specifically to markers of aging in immunity, and it is not clear whether they are informative about changes in other tissues that may not be harvested in wild primate populations.

We have revised the text to emphasize that our analyses pertain specifically to methylation in peripheral blood, so may be more relevant to immune function than if we had conducted them in a different tissue (lines 64; 109-112).

Another, related point is the fact that the functional significance of epigenetic clocks (why they "work" at all) remains uncertain. The extent to which they are in the causal pathway to health outcomes (as suggested by at least one recent study that likely postdates this draft: Lu, Yuancheng, et al. "Reprogramming to recover youthful epigenetic information and restore vision." Nature 588.7836 (2020): 124-129), or merely a marker, is unclear. In the end, these are useful measures if they predict relevant outcomes, but I think noting this current ambiguity around their functional basis might warrant a sentence or two in the discussion.

We have revised the text to highlight that the functional significance of DNA methylation-based epigenetic clocks—i.e., whether they are passive biomarkers or part of the actual mechanisms of aging—remains unclear (lines 302-304).

Previous Reviewer #4

Anderson et al. have produced a manuscript investigating DNA methylation (DNAm) assessed epigenetic age in a wild baboon population in Kenya. They have constructed a clock from RRBS DNA methylome data from whole blood from 245 baboons (277 samples). This clock comprised 573 CpGs and could predict age within a mean absolute difference of 1.1 years ± 1.9 SD (r = 0.762). The accuracy of this clock was subsequently compared against other age- associated traits and between sexes. It tracked well with age-related phenotypes, such as molar dentine exposure, but was less accurate in females. This divergence occurred after 8 years when the majority of males achieve adult dominance rank and disperse from their birth group. Clock-assessed age acceleration was not associated with the strongest predictors of lifespan in this population, early adversity and social integration. Instead, male dominance rank was most strongly associated with positively accelerated age.

This manuscript has already been reviewed, so the responses to this first round of reviews are available, and as well, I have assessed the methodological approach. The authors’ interpretation of these results is a ‘life fast die young’ effect. My concerns for the authors to consider are listed below.

Major

Thanks for making this point. We have revised this text to emphasize that it is the first clock to assess the effects of natural variation in the social environment in wild primates (lines 74-75). This change emphasizes the most important aspect of our study rather than the construction of the clock itself.

2. Regarding the composition of the clock (line 104), it is incorrect regarding human clocks being specifically enriched in genes, CpG islands, promoter regions, and putative enhancers, compared to the background. These clocks have been devised with data from human DNAm arrays, which predominately only probe these regions – the earlier 27k array used for the construction of the Horvath clock is almost exclusively promoter based. It is stated that the baboon clock is functionally important for gene regulation, which is actually not specifically the case for human clocks [3].

We have clarified that the enrichment observed in humans was relative to the background set of sites tested (i.e., measurable CpG sites on the array) rather than the background of all CpG sites in the genome (lines 95-97). Similarly, in our analyses, we performed enrichment analyses relative to all sites that could have been possibly included in the clock (i.e., those profiled by RRBS; lines 429-430), not all CpG sites in the genome (RRBS analysis also enriches for a subset of functional compartments, especially promoters and CpG islands, compared to all sites in the genome).

3. What was the explanation of the outlying result for the Dominance rank 10 in Figure 2? – How robust are these ~20 ranks – as seems a large number?

We do not have a definite explanation for the apparent increase in relative epigenetic age in rank 10 animals relative to the fit linear relationship. We note that this outlier is based on only 5 individuals, so it may simply reflect stochastic sampling. Indeed, 2 of the 5 males have relative epigenetic age values below 0, so the overall pattern is driven by only 2 other males with large positive values. None of these males are unusual in terms of read depth or age at the time of sampling, and none experienced major shifts in rank close to the time of sampling.

With respect to our rank assignments, most groups have smaller numbers of post-reproductive maturation males, but some groups in Amboseli range up to 120 animals and have more males (and hence, more positions in the hierarchy). Our methods for assigning rank are based on the same near-daily observations used for this population to assess male rank effects on mating success, paternity success, glucocorticoid levels, lifespan, and other outcomes (see lines 519-523). Thus, we view them as generally highly reliable, although in periods where there is a high degree of rank competition, there can be rapid change.

4. The authors state that Amboseli α males also exhibit elevated glucocorticoid levels, increased expression of genes involved in innate immunity and inflammation. Is there any support in the DNAm data for this in the high-ranking males – i.e. showing epigenetic changes also found in human blood consistent with this? [4, 5].

Thanks, this is an interesting question that is the subject of ongoing analysis in our group. Specifically, we are investigating the socioecological predictors of DNA methylation at individual CpG sites, as well as the potential for this variation to explain earlier published results on dominance rank and gene expression (Lea et al. 2018, *PNAS*). We view these analyses as beyond the scope of the current manuscript. However, we do show that the clock sites that contribute to male rank effects on relative epigenetic age are enriched in or near genes important in the gene expression response to the pro-inflammatory agent lipopolysaccharide (lines 102-109; Figure 1 —figure supplement 3).

5. There is significant complexity in the baboon genome [6]. What is the potential impact of the use of the mixture of different Papio species – cynocephalus and anubis? Were the CpGs in the clock assessed for any evidence of genetic confounding in the two separate species – by methods such as Gap Hunting? [7]

Thanks, we have also worried about possible mapping (and hence DNA methylation quantification) errors introduced by the hybrid nature of our population, especially because the reference genome is for the anubis baboon (we note that we did not use a mixture of individuals from different species here; rather, all of the animals in Amboseli are hybrids between these two ancestries: see Wall et al. 2016, *Molecular Ecology*). We therefore addressed this concern in a previous paper (Vilgalys et al. 2019, *Molecular Biology and Evolution*; see Supplementary Information 1.5 and Figure S7). Specifically, we simulated bisulfite sequencing data from both the anubis genome and the yellow baboon genome and quantified mapping and quantification biases depending on the ancestry of the original sample. In brief, we found that the percent of incorrect calls differed by less than 0.4% between yellow and anubis baboons, making it highly unlikely that they affect our results. We reference these previous results in the revised text (lines 382-387). Notably, if CpG methylation measures were strongly affected by ancestry in our data set, those sites would also become poor predictors of age and would be unlikely to be incorporated into the epigenetic clock.

6. Was any evidence or adjustment made for potential batch effects as the RRBS was a combination of two analytical batches? (n=36 and n=277).

We did not perform batch correction prior to creating the clock (although we do so routinely when analyzing DNA methylation variation at individual CpG sites: e.g., Lea et al. 2015 *PLoS Genetics*). This choice was made because we were interested in developing a generalizable clock that could predict well across batches; additionally, correcting for batch effects in the full sample could “leak” information about the full data set into the test samples. However, we confirmed that our results in the main model (Table 1) were robust if we added batch effect (previously generated samples [n=36] versus newly generated samples [n=241]) as a covariate. We have clarified these points in the revised paper (lines 411-416).

7. With all the caveats and issues with BMI, sex-specific effects etc the authors need to confirm that data quality issues have not impacted on the conclusion drawna. Specifically, there is a big range of sequencing depth 17.8 (± 10.5 s.d.) million reads. 20 Million reads normal for good (www.diagenode.com/en/p/rrbs-service).

Thanks, we agree that deeper read coverage can provide better estimation of individual CpG site methylation levels. However, we observe no effect of read depth on ∆_age_ in our data, in either males or females. Specifically, we checked whether including an additional covariate of read depth in the models reported in Table 1 explained additional variation in ∆_age_. It did not (p = 0.11 for females and p = 0.23 for males), and also did not change any of the other parameter estimates. We have therefore elected to keep the original model results in the main text, but revised the text to note that our results are robust to variation in read depth (lines 213-216; Supplementary Table 5).

b. There is a negative correlation between age and reads that is stronger in males

Thanks for pointing this out. We now highlight in the revised text that, although read depth is correlated with chronological age in our sample, read depth is not correlated with ∆_age_, the value of interest in our analysis (lines 213-216; see also our response above). Importantly, we also control for chronological age in our analysis of ∆_age_.

c. A threshold of 35x is low, 10x more standard (ENCODE) – “at least 10-fold coverage of a CpG is required for accurate measurement of percent methylation” https://www.encodeproject.org/documents/e82fdfdf-f387-47d8- af03-df67bbea0e72/@@download/attachment/2010-05-30_mod- ENCODE_TF_Chrom_Data_Standards.pdf 30x used for Roadmap standard – and even issues at this level [8]

We agree that 5x coverage will translate to some noise in DNA methylation level estimation for any given CpG site. However, unlike the Roadmap Epigenomics data sets, we do not intend for any particular sample to serve as a reference data set for the field. Rather, we are focused on understanding the sources of population-level variation across hundreds of samples, with a primary interest in composite prediction accuracy across the methylome, rather than at individual CpG sites. Thus, our data reflect a classical trade-off between analyzing more individuals versus sequencing each individual at deeper coverage. In population-based studies, favoring more individuals is a common solution.

d. Coverage issues are the clear disadvantage of sequencing-based clocks – with much more stochastic coverage of the CpGs included. This further reduces its use in any replication compared with the human data using robust and consistent CpG arrays [3]. How well were the 573 CpGs actually chosen in the clock covered in the cohort? Age changes are ubiquitous so multiple clocks should be able to be constructed from these data. If only build the clock (excluding these 573) from a smaller number of higher confidence CpGs from the ~458k with higher CpG coverage are results consistent?

We agree that multiple clocks can be constructed from the data. Indeed, our clock incorporates all 573 unique sites that were used in at least one epigenetic clock in our training-test sets. Median average coverage for sites included in the clock was 23.0x. Alternative clocks can be built with similar prediction accuracy and smaller numbers of sites, e.g., by increasing the strength of regularization (α parameter; Figure 1—figure supplement 2).

e. There is a small distinct number of outliers for BS conversion (lower ~98%) – which could also impact particularly as the wild-card aligner BSMAP is more susceptible to false positives related to inefficient conversion [9].

We agree that small differences in conversion can lead to artifacts (although no bisulfite conversion rates were < 98%). To address this concern, we checked whether the ∆_age_ models in Table 1 were robust to exclusion of the 6 samples with bisulfite conversion rates < 0.99. All results are qualitatively the same, and all significant results remain significant. We obtain the same result if bisulfite conversion rate is run as a model covariate. We have revised the text to note that our models are robust to the small variation in bisulfite conversion rates (lines 213-216; Supplementary Table 5).

There was a higher variance in females – were sex chromosome CpGs excluded from clock? Were the enrichments compared against the background set of the possible ~458k CpGs these clock CpG came from?

Yes, we constructed the clock based on autosomes only, and all enrichments were relative to the background set of 458k sites that we originally used in calibrating the clock. Both points are now clarified in the text (lines 381 and 93-97, 429-430).

State the tissue type used for the ENCODE H3K4me1 ChIP-seq data from humans that was lifted over.

The tissue type was peripheral blood mononuclear cells (PBMCs), now noted in line 438.

References:

1. Horvath, S., DNA methylation age of human tissues and cell types. Genome Biol, 2013. 14(10): p. R115.

2. De Paoli-Iseppi, R., et al., Measuring Animal Age with DNA Methylation: From Humans to Wild Animals. Front Genet, 2017. 8: p. 106.

3. Bell, C.G., et al., DNA methylation aging clocks: challenges and recommendations. Genome Biology, 2019. 20(1): p. 249.Ligthart, S., et al., DNA methylation signatures of chronic low-grade inflammation are associated with complex diseases. Genome Biology, 2016. 17(1): p. 255.

4. Tang, R., et al., Adverse childhood experiences, DNA methylation age acceleration, and cortisol in UK children: a prospective population-based cohort study. Clin Epigenetics, 2020. 12(1): p. 55.

5. Rogers, J., et al., The comparative genomics and complex population history of Papio baboons. Sci Adv, 2019. 5(1): p. eaau6947.

6. Andrews, S.V., et al., "Gap hunting" to characterize clustered probe signals in Illumina methylation array data. Epigenetics Chromatin, 2016. 9: p. 56.

7. Libertini, E., et al., Saturation analysis for whole-genome bisulfite sequencing data. Nat Biotechnol, 2016: p. doi:10.1038/nbt.3524.

8. Rauluseviciute, I., F. Drablos, and M.B. Rye, DNA methylation data by sequencing: experimental approaches and recommendations for tools and pipelines for data analysis. Clin Epigenetics, 2019. 11(1): p. 193.

[Editors' note: we include below the reviews that the authors received from another journal, along with the authors’ responses.]

Reviewer 1In this paper, Anderson et al. have claimed that high social status have been found to accelerate the epigenetic aging rates in wild baboons. However, in the sample set most of the highly ranked animals had higher BMIs. Higher BMI has previously been reported to increase epigenetic age. Therefore, it is unclear to me how novel the authors’ claims are. Overall, this study is not conclusive, and additional studies should be considered for this study to be relevant and accurate.

Thanks for this clear summary of your major concern. In response, we have conducted a series of additional analyses, which increase our confidence that BMI does not drive the results we report for male rank.

Specifically, we now show that if we force rank and BMI to be independent by modeling Δ_age_ as a function of the residuals of male dominance rank controlling for BMI (instead of dominance rank itself), residual dominance rank continues to significantly predict Δ_age_ in the same direction as our original model (p_rank(residual for BMI)_ = 9.95 x 10^-4^; lines 198-200; Supplementary Table 5). In contrast, if we model the residuals of BMI controlling for dominance rank, BMI does not significantly predict Δ_age_ (p_BMI(residual for rank)_ = 0.139), but dominance rank remains highly significant (p_rank_ = 1.88 x 10^-4^; lines 198-204, Supplementary Table 5). These results are consistent with our original finding that male rank significantly predicts Δ_age_ when BMI is included as a covariate (lines 191-192). We note that in our main analysis we modeled BMI controlling for chronological age (now denoted as age-adjusted BMI for additional clarity, lines 172-173), resulting in a measure of BMI that is uncorrelated with dominance rank (Pearson’s *r* = -0.068, p = 0.516; see lines 167-172 and Supplementary Table 4). However, this decision does not affect our results: in an alternative multiple regression that uses raw BMI, dominance rank is still a significant predictor of Δ_age_ but raw BMI is not (likely because variation in raw BMI primarily captures growth and development in baboons; lines 198-204).

These results, now reported in lines 196-204 and in the Supplementary Materials (Supplementary Table 5), dovetail with our finding that age-adjusted BMI also does not predict Δ_age_ in female baboons (Table 1). Below, we also discuss the difference between the interpretation of variation in BMI in humans and variation in BMI in wild baboons, which—along with the new analyses described above—make it highly unlikely that our results simply recapitulate previous findings about BMI and epigenetic aging in humans.

Major comments:1. It is well known that a higher BMI is associated with accelerating epigenetic aging and this has been shown in previous reports from multiple research groups (PMID: 30785999, PMID: 28289477, PMID: 29159506, PMID: 28089957, PMID: 31001624, PMID: 31480455, PMID: 28198702). In this study, there seems to be a trend for higher ranked animals having a higher BMI. With a higher social hierarchy, these animals would have a higher priority to accessing foods and consequently would have higher BMIs. It cannot be denied that the detected phenomenon potentially is only reflecting the BMI differences in the epigenetic age, not necessarily their social ranking. Therefore, it is not surprising at all that there is a correlation between acceleration of epigenetic aging and social rank because the higher ranked animals have higher BMIs.I am not sure whether the authors’ finding has any novelty. I believe the actual novelty of this study is in fact that higher social ranked baboons had higher BMIs. Rather than the effects of ranking on epigenetic age, it would be interesting if the authors focused on the differences between sexes found in the relationships of rank and BMI. In males, higher ranked animals have higher BMI. In contrast, lower ranked female animals have relatively higher-BMI, although higher ranked (1-5) animals have high-BMI.

Thanks, these comments have led us to clarify some important distinctions between the variation observed for BMI in humans (the subject of all the studies cited above) versus the variation in BMI observed in wild baboons. Specifically, in the human-centered studies above, high BMI translates to classification as overweight or obese. In contrast, wild baboons in Amboseli are never overweight or obese: females average 1.9% body fat, based on subscapular skinfold estimates, and males average <9% body fat, based on stable isotope measures (skinfold estimates are unreliable in males) (Altmann et al. 1993, *American Journal of Primatology* 30: 149-161). These values, which are several times lower than typical body fat percentages for obese men and women, reflect the fact that BMI variation in male baboons is a function of lean muscle mass, a distinction we now clarify in the revised manuscript (lines 166-167; 196-197; 286-290).

Our findings are thus distinct from the epigenetic age-BMI associations in the papers cited above. Additionally, based on the new analyses described above (lines 196-204), we are confident that our findings are driven by rank, not BMI, although age-adjusted BMI has some additional predictive value for Δ_age_ in males after rank is taken into account (Table 1). We now clarify this point in the revised manuscript (lines 286-292).

With respect to novelty, we also clarify that our results report the first epigenetic clock to be calibrated for a wild primate (as noted by Reviewer 2) and are the first to establish a link between social factors and epigenetic aging in any natural animal population (lines 72-76). Finally, with respect to sex differences in the rank-BMI relationship, there is no relationship between age-adjusted BMI and rank in the females in our sample (Pearson’s *r* = 0.058, p = 0.646). This is not surprising because rank in females is nepotistically determined, but rank in males is determined by physical competition (see Hausfater et al. 1982, *Science* 217: 752-755 and Alberts et al. 2003, *Animal Behaviour* 65: 821-840). We have thus chosen not to focus on sex differences in the BMI-rank relationship because this difference in rank dynamics is well-established in the literature. However, your comments stress the importance of providing this context to readers, and we now do so in lines 304-307.

2. The authors should make clear the universal effects of social ranking in other sample sets to show whether the detected phenomena is specific to this sample set/cohort/species. For example, there is methylome data derived from age matched rhesus monkey samples with social rank information in a previous publication (PMID: 22493251). The authors could analyze the epigenetic age differences between high and low ranked rhesus monkeys to compare (and contrast?) with their gathered data from their sample set.

Thanks for the suggestion. Our results strongly suggest, which we now clarify, that rank effects are in fact *not* universal (lines 191-192; 313-314). Specifically, we argue that the observed sex-specific rank-Δ_age_ association stems from the nature of dominance rank determination in wild male baboons, where rank is dynamic and linked to physical competition. In contrast, rank is highly stable and nepotistic in both female baboons and female rhesus macaques (who are the subject of PMID: 22493251). Indeed, one hypothesis generated by our findings is that high dominance rank specifically accelerates epigenetic aging when achieving and maintaining rank is energetically costly. The degree to which this condition holds differs across species, and, within species like baboons, also between males and females (the energetic costs of rank competition are much more relevant in males). We have highlighted this point more effectively—including the predictions it makes about when and where our findings should generalize—in the revised manuscript (lines 313-321).

Notably, the study cited above (PMID: 2249325, which was led by the senior author of the present study) used experimental manipulations to randomize dominance rank in captive female macaques; it also generated quite limited data on DNA methylation (N = 6). Thus, it does not capture natural rank dynamics for baboons (or rhesus macaques) of either sex. Because it is not an appropriate comparison case, and no other research group has generated population-level DNA methylation data for wild primates, we have now revised the Discussion to include promising test cases for future work (lines 316319).

3. To eliminate the effects of the differences in BMI and chronological age, the authors should only assess animal samples from same age ranges and BMI, and further analyze their agerelated methylation status by dividing into high and low social rank. However, it seems like the number of samples matching these conditions are short of analyzing with enough statistic power. As the authors pointed out in the Discussion section, additional longitudinal sampling will be necessary to make the conclusion the authors have made.

Please see our responses above and lines 196-204, where we report a series of analyses to address the concern about confounding between rank and BMI.

4. The authors tried to detect the effects of promotion/demotion in social ranking (figure 3). For their claims that social ranking has effects on epigenetic aging speed, the authors failed to test using BMI, which is another significant covariate. Without doing so, it is not possible to eliminate the possibility that BMI status affects epigenetic aging speed. I wonder whether the BMI changes (samples collected at higher/lower BMI) is correlated with epigenetic aging (relative epigenetic age).

Thanks for this suggestion. We have now performed an analysis of whether longitudinal change in age-adjusted BMI predicts longitudinal change in Δ_age_, which parallels our analysis of longitudinal change in male rank. We find that both change in age-adjusted BMI and change in dominance rank significantly predict change in epigenetic age, although the effect of rank is larger (lines 254-255). Consistent with this observation, longitudinal change in the residuals of rank after regressing out raw BMI still explains an estimated 20% of the variance in longitudinal change in Δ_age_, although this effect is no longer significant. In contrast, BMI adjusted for rank explains almost none of the variance in change in Δ_age_ (R^2^ = 0.01) (lines 255-259). These results again suggest that our findings are a consequence of rank, not BMI.

Minor comments:1. Use same scale both in females and males in Figure 1 to provide better comprehendible visualization. Add dashed lines for y=x in Figure 1C and 1D.

We have made these changes (see revised Figures 1C-D).

2. Need to change the title of Figure 2 as appropriate. It is too strongly worded and does not seem to reflect the contents shown in the figure.

We have revised to “Dominance rank predicts relative epigenetic age in male baboons,” which directly corresponds to the y-variable we plot in Figure 2.

3. Add unit (years) for “Age” in the Supplementary Table 1.

Done.

4. Plot and check the association between age (x axis) and methylation percentage (y axis; for example, average of each individual in each genetic regions such as promoter, CpG island, shore etc) with color codes based on ranking (for example, by grouping; 1-5, 6-10, 11-15, etc) using CpG sites showed age-related increase/decrease in DNA methylation.

We have included this information in new Supplementary Figure 5, based on CpG sites included in the epigenetic clock. Change with age is visually apparent for clock sites in all genomic contexts.

5. Include epigenetic age (biological age) in the Supplementary Table 1.

Done.

6. Include genomic region of the CpG site with gene name in the Supplementary Table 2.

Done.

7. Correct the reference figure (Figure 3 to 4) in line 199, p7.

Done.

8. Perform further analyses about age-related methylation changes in wild baboon such as volcano plot between young and old, then pathway assay using significant genes etc. in each sex.

Thanks for the suggestion. We performed site-by-site age-associated differential methylation analysis, identified age-related differentially methylated regions, and performed pathway enrichment analyses in a previous paper (Lea et al. 2015, *PLoS Genetics* 11: e1005650). Consequently, we have not repeated these analyses because they are somewhat tangential to our main point, given limited space. However, we now report site-level analyses for the 573 sites in the calibrated clock (Supplementary Figure 4; lines 94-99). Additionally, we have made the complete DNA methylation data set and age/sex information publicly available to support further re-analysis (newly generated data are available via NCBI Sequence Read Archive project accession number PRJNA648767).

Reviewer 2:This is an interesting manuscript which does a lot – both building and validating an epigenetic clock in the Amboseli baboons, and then looking to see which factors predict deviations in epigenetic age relative to chronological age. There’s lots to like about this study, and it’s perhaps (?) the first published epigenetic clock from a free-ranging primate population. I think there’s some components of the manuscript that could be improved.

We appreciate the positive feedback, as well as the constructive suggestions for improvement.

My major comments are:1. The manuscript needs structure. There is not a single subheading in the long “Results and Discussion” section, which keeps going and bringing up new issues and new analyses. The lack of a macrostructure makes it much harder to read and follow than it needs to be. I think the most useful thing would be to set out a list of (numbered?) aims/objectives in the last paragraph of the Introduction, so it’s clear what the specific set of aims is. For example, this could be something like: 1) Epigenetic clock validation; 2) Sex differences; 3) Predictors of epigenetic age. The Aims should then be used to structure the manuscript as subsections of the Results/Discussion. It will make it easier to follow.

Thanks. We have followed your suggestion to lay out the study aims in the introduction, and have added subheadings to delineate natural breaks in the results. We also separated the Results and Discussion sections to improve readability.

2. One of the major contributions of the manuscript is the development and validation of the epigenetic clock. I think it’s likely that this will be used in many further Amboseli papers with the current manuscript referenced for the validation, so I think it’s important to make sure that this description is as useful as possible. In that regard, I’d like to see more information on how well the 593 chosen CpG sites predict chronological age specifically compared to other potential combinations and numbers of sites.

Thanks for this suggestion. We now include additional information in the text (lines 8891; 371-375) and a new supplementary figure (Supplementary Figure 2) that describes how we identified the 573-site clock that we applied in the main text. In brief, we used a standard approach for the elastic net to identify the optimal trade-off (α) between L1 (Lasso) and L2 (ridge regression) regularization. The resulting set of sites includes a core set of sites that are always chosen as part of the clock, regardless of α, as well as a smaller number that are more variable. In new Supplementary Figure 2, we also show how all 573 sites perform, in terms of median absolute difference and R^2^ between predicted and true chronological age, compared to more restrictive sets of CpG sites.

Line 82 – also please give a measure of variance around the difference between the chronological age and that predicted from the epigenetic clock.

We now report the standard deviation of this difference in line 86.

In addition, the authors state: “The predicted ages from these 95 longitudinally collected samples were older for the later-collected samples, as expected (Figure 1C-D; binomial test p = 5.95 x 10-5 96 ). Furthermore, the change in epigenetic clock predictions between successive longitudinal samples positively predicted the actual change in age between sample dates (β = 0.312, p = 0.027).” Is it possible to provide more information here? What is the mean (with a measure of variance) difference between the amount of intra-individual chronological aging and that predicted by the two measures in the epigenetic clock?

We now report the mean and standard deviation of the difference between change in chronological age and change in epigenetic age predictions in line 121.

3. I don’t have a good sense of what the Authors think the mechanism is by which male dominance rank impacts the pace of aging, nor other elements of genomic regulation and downstream physiology. In part I think I’m unclear how they see dominance rank itself. Do they see it purely as a behavioral construct that is made up of different behavioral measures, and which simply reflects patterns of e.g. aggression given and received, and the associated behaviors intended to mediate that aggression (displacement, avoidance, etc.). Or, do they think that dominance rank reflects something separate and independent, which we are able to *measure* behaviorally, but which exists separately to those behaviors. If the former, then perhaps the behavioral measures themselves are the best thing to model rather than dominance rank itself? In this regard a specific question: does dominance rank predict epigenetic age better than, say, aggression given to other adult males, aggression received from other adult males, or other behaviors that primarily determine an individual’s assigned dominance rank? I would like to see such analyses. To be clear: I’m not sure if the Authors think that it is the behaviors themselves, such as giving and receiving aggression, that lead to changes in the genome, or something else intrinsic about “high social status” that they are capturing via their dominance rank measure. If what ages high ranking males faster is the constant aggression involved in being challenged and putting other individuals in their place, then in my view it is better just to focus on the behaviors involved as the mechanisms linking behavior to the body’s biology and condition are much clearer.

Thanks for this very interesting and important set of questions. We feel that we are limited in being able to address them here, primarily because our ability to assign male dominance rank is substantially better than our estimates of the rates of agonistic interactions that underly these ranks. Specifically, our rank assignments depend upon sampling enough agonisms to establish *consistency* in the patterns of wins and losses among males. In contrast, to test whether dominance rank is a *better* predictor of outcomes than rates of agonistic interactions would require highly accurate measures of rates. We don’t currently sample male agonistic behavior at a sufficient level of intensity to make us confident in such a comparison; male-male interactions can be subtle and occur very quickly (e.g., displacements, threat faces), and male agonism rates can vary substantially over short periods of time (e.g., males can be relatively pacific until there is active competition for rank—and we can miss those episodes if the group is not observed on the corresponding days). Thus, we believe the rank data are likely to be systematically more accurate than estimates of agonism rates themselves.

Indeed, when we asked whether substituting male dominance rank for either agonisms given or agonisms received is a better predictor of Δ_age_ than rank itself, we found (i) that only agonisms given was a significant predictor of Δ_age_; but that (ii) a model including rank is a better fit to the data than a model substituting rank for agonisms given (agonisms directed to any group member: Δ_AIC_ = 7.11; agonisms involving only adult males: Δ_AIC_ = 3.13). We also found that agonisms given is not a significant mediator of the rank-Δ_age_ relationship (p = 0.053); further, although the trend is suggestive, the mediation effect only attenuates the original rank effect by a small percentage (19%). These results are consistent with two, non-mutually exclusive explanations: first, that rank captures something beyond the effects of agonism rates alone; or, second, that measures of agonism rates are simply more noisy than measures of rank. Because we are not able to differentiate these explanations here, we have chosen to leave these analyses out of the revised manuscript.

However, we agree on the importance of your question about the “meaning” of rank. While fully addressing it is beyond the scope of this study, we do now discuss our results in the context of previous work in the Amboseli baboons, which was able to deploy a Mendelian randomization (MR) analysis to investigate the directionality of rank-gene expression associations in the Amboseli baboons (Lea et al. 2018, *PNAS*). In that case, MR analysis suggested that, instead of being a *consequence* of rank, gene expression differences preceded attainment of high rank. Those findings suggest that the male rank epigenetic age relationship we describe in this paper may capture more about the characteristics of males who compete successfully for high rank than about the behaviors they exhibit upon attaining it (now discussed in lines 292-299).

Lastly, I also think that given the centrality of male dominance rank to the manuscript’s conclusions, more details on how this is measured and calculated would be useful, including more details on the specific behaviors included.

We have included additional information in the Methods about how dominance rank is measured and calculated in the Amboseli baboons (lines 474-482).

4. The absence of certain effects seems to be as interesting as the presence of the rank result. Recent manuscripts from these researchers and this population have shown that early-life adversity is extremely important in shaping a variety of measures of biological condition and life history. Some of the discussion in the present manuscript does consider potential reasons why there were no such effects here, but I thought more could be done – as the Authors state, it’s surprising given their prior results. I also felt that the significance of the absence of other effects on epigenetic age could usefully be discussed – for example, nothing is really made of the fact that female rank does not predict epigenetic age. As the Authors note, the result presented here seems most similar to the results found on GC concentrations and male dominance in this study population. What do the present results tell us about how aging effects can operate differently within and between biological domains, and about the interaction between different elements of allostatic load? The Amboseli Baboon Project has published a large number of studies relating social behavior and social status to measures of health and fitness. For those of us on the outside trying to understand the overall picture, it would be useful to read more explicitly how the authors feel that the present results fit together (or not, where appropriate) with their other findings involving other measures and markers.

Thanks; we agree that the absence of significant associations is interesting in and of itself. In response, we now discuss two themes emerging from work on the Amboseli baboons (lines 262-283; 303-321).

First, dominance rank effects on physiological and molecular outcomes—epigenetic aging, in this paper, and also immune gene expression (Lea et al. 2018, *PNAS*) and glucocorticoid levels (Levy et al., in review at *Hormones and Behavior*; Gesquiere et al. 2011, *Science*)—are consistently more detectable and/or stronger in males than in females. This difference likely arises from major differences in how rank is attained and maintained in male versus female baboons (lines 303-321). Second, major predictors of lifespan (e.g., cumulative early adversity and social integration, in this population) are not necessarily major predictors of physiological and molecular measures. This may be because they act through entirely distinct pathways, because their effects are tissue, cell type, or environment-specific, or because they are sensitive to specific types of early adversity, but not others. Further research will be necessary to tease apart these possibilities (lines 262-283).

Reviewer #3:Anderson et al. have developed a “DNA methylation-based age predictor”, i.e. an epigenetic clock, for the wild baboons at Amboseli, Kenya and they find that male dominance rank is associated with significant differences between predicted age and chronological age. The study is well-designed and well-executed, and the paper is well-written and a pleasure to read. The results are fascinating and have broad relevance for aging in humans and other non-primates as well as addressing bigger questions of evolution. It’s fascinating that neither cumulative early life adversity nor social bond strength explained variation in the difference between predicted and chronological age, but male dominance rank did. Higher social status in males is associated with more rapid epigenetic aging even though higher social status yields increased resource access and is generally associated with a better body condition, higher fertility and better competitive advantage. These results are novel and only possibly in a population like the Amboseli baboons that have been studied for so long with a wealth of data available on each individual. Specifically, this study provides new results comparing the impact of different psychosocial stressors on epigenetic age and furthermore suggests that the impact of epigenetic aging lasts only as long as the stressor exists.

Thank you.

The paper should be published and I have only a few comments:– Could the authors speculate on why their clock was more accurate in males than females – sample sizes are comparable so it must be something else. The faster rate of biological aging in males relative to females would seem to suggest that the clock would be less accurate, not more accurate, in males than females.

Thanks for this question. Given that female baboons live longer than males (Colchero et al. 2016, *PNAS*), we think it is perhaps unsurprising that they show slower epigenetic aging age than males in adulthood. Notably, if we fit a clock only for animals up to age 8, when the slopes for males and females diverge, they are similarly accurate (lines 130133). Thus, the relative flattening of the slope in older females likely accounts for better overall prediction in males relative to females (although we cannot exclude some degree of viability selection in females). We now explain this interpretation in lines 146-149.

If choosing different CpG sites would create a better epigenetic clock for females, what does it mean that different CpG sites are methylated differently in response to aging in males vs females?

We believe that the observed sex differences in clock performance reflect changes that occur at the same CpG sites, but with higher variance in females. Indeed, if we estimate the effect of age on DNA methylation for the 573 clock sites in males versus females, these estimates are well-correlated between sexes (Pearson’s *r* = 0.91, p = 3.35 x 10^204^), but are estimated with more certainty, per site, in males than in females (e.g., the standard error are systematically smaller in males; now reported in lines 143-146 and new Supplementary Figure 4).

– It was interesting to read about the complexities of male high rank and age, i.e. high rank exists primarily between 7-12 years of age whereas low rank exists across all ages.Furthermore, it was fascinating that lower rank consistently associated with lower predicted age even when the lower rank was in a formerly high-ranking male, e.g. two males who were sampled later in life when they were of a lower rank both showed a decrease in predicted age, despite greater chronological age. This suggests that the increased epigenetic aging due to high rank is not a permanent change, i.e. does not result in increased aging across the lifespan but just during the tenure of high rank. The implications for the ‘recency’ vs ‘accumulation’ model are highly relevant. I look forward to seeing future work by Tung’s group with more longitudinal samples.– The methods used are robust and the methods section, as well as the results, are very detailed and well-written.

We appreciate the positive feedback!

– The Results and Discussion section ends abruptly with the discussion of BMI and the costs associated with investment in body condition. I suggest a concluding paragraph that brings the reader back to the results on male dominance rank.

We have restructured the results and discussion based on comments from Reviewers 1 and 2, and now discuss BMI-related results in lines 284-302. The revised manuscript now concludes as suggested, by focusing on the implications of our results for understanding the consequences (and potential costs) of dominance rank in different types of hierarchies.

– I could not find project PRJNA607996 in the NCBI SRA but I assume that’s because access is limited until the paper is published.

Yes, that’s correct. However, a reviewer link is available: https://tinyurl.com/y5xuhdxf.

Response to second decision letter

Reviewer #1:I do not agree with the author’s response about the correlation between age-related methylation status and BMI. In humans, even in the cases with normal BMI (<25), there were tendencies of accelerations of epigenetic aging as the BMI increased (at least several reports; PMID: 28289477, PMID: 28089957, PMID: 25313081, however it would depend on sample sets). I assumed that there would be the possibility of this variation existing in wild baboons as well.

Thank you for this perspective and the additional references. We highlight two important points that support our original response: specifically, that the effect of BMI on epigenetic age in male baboons is distinct from the epigenetic age-BMI associations in published human studies.

First, we explicitly test the hypothesis that BMI predicts Δ_age_ (our measure of biological age: predicted age – chronological age). We report that male baboons with higher age-adjusted BMI do appear somewhat old-for-age, although only when dominance rank is accounted for (lines 223-224, 230-232, 330-332; Table 1 and Table S5). This pattern is only observed in males: there is no evidence, in any of our analyses, that BMI predicts epigenetic age in baboon females (lines 223-224; Table 1).

Second, all of the papers cited above include individuals that are overweight and/or obese, and none perform analyses on subsets of the data that exclude those individuals. Indeed, two of them explicitly focus on the effect of obesity in their titles (PMID: 28289477, “Obesity accelerates epigenetic aging in middle-aged but not in elderly individuals”; PMID: 25313081, “Obesity accelerates epigenetic aging of human liver”). Because these papers focus on the range of BMI variation in developed human populations, we do not feel that they provide strong enough priors to override the actual data analysis performed in our study.

However, the reviewer’s comment brings up an important caveat that we now clarify in the text—BMI in humans is calculated differently from how we calculate BMI in baboons (baboon BMI = body mass divided by the square of *crown-rump length*; human BMI = body mass divided by the square of *height*; our approach follows the precedent set in Altmann et al. 1993, *AJPA*; Altmann et al. 2010, *ANYAS*; and citations therein). Thus, the scales of BMI are species-specific, such that a BMI of 45 in humans represents extreme obesity, but a BMI of 45 in baboons represents a healthy weight. To ensure that readers do not default to expectations set by human studies, we have placed additional emphasis on the distinction between baboons and humans in the revised manuscript, including explaining the difference in BMI calculation and noting that body fat percentages are very low in Amboseli animals (lines 189-190, 334-338, 570-573).

However more importantly, according to this study, I noticed an acceleration of epigenetic aging in the lower BMI group (paraphrased as young age group) as well which I personally have never seen before, despite the significant correlations in BMI/rank and BMI/age. The lowest BMI group included animals with lower rank which cannot be explained by their claim of aging accelerating in relation to the rise in rank. It seems like that the aging acceleration occurred not only in high-ranked animals but in low-ranked animals as well. The trend showing that the younger animals have a higher epigenetic aging speed is visible in Figure 1B in the manuscript when checked carefully. The fact cannot be ignored and I believe the authors should change the title of this manuscript.

Male baboons do not achieve full adult body size until several years after reproductive maturation (i.e. testicular enlargement). This produces a correlation between raw BMI and chronological age, as we show in new Supplementary Figure 7 (note that the correlation emerges entirely from having younger males, including a small number of pre-maturation males [n = 11] in the sample). Δ_age_ (predicted age – chronological age) is also correlated with chronological age because the epigenetic clock tends to overestimate the ages of young baboons and underestimate the ages of old baboons (as shown in Figures 1A-B and explained in lines 205-212). We now note that this compression effect has precedent in epigenetic clock and elastic net regression analyses (see for example Levine et al., *Aging*, 2019; Engebretsen and Bohlin, *Clinical Epigenetics*, 2019]), including some of the foundational work on epigenetic age prediction (the “Hannum clock:” Hannum et al., *Molecular Cell*, 2013; lines 208-212).

Together, the raw BMI-chronological age relationship and the Δ_age_-chronological age relationship produce the patterns reported by the reviewer, which were based on simple bivariate correlations. However, these spurious patterns are eliminated in our multivariate linear modeling approach because we explicitly control for chronological age as a covariate in our linear models. Indeed, if we do not correct chronological age in this manner, *any* variable that is correlated with chronological age will predict Δ_age_ (see simulations that demonstrate this pattern, in response to the next reviewer comment).

Nevertheless, the comments above motivated us to perform a fifth additional analysis to check whether our findings are confounded by BMI (new results in lines 232-235, complementing four other alternative checks; Supplementary Table S5). We now show that if we drop all low BMI samples from our analysis (BMI < 41 and 31% of our data set for males, chosen to eliminate all males who clearly have not completed full growth; see retained points above the dashed line in Author response image 2), we eliminate the correlation between BMI and Δ_age_ altogether in the remaining sample (Pearson’s *r* = 0.04, p = 0.67; Author response image 2). However, eliminating low BMI/young males does not change any of our findings about the relationship between male rank, age-adjusted BMI, and epigenetic age (rank effect pvalue = 7.14 x 10^-3^ in the reduced data set; Author response image 2; model now presented in Supplementary Table S5). This result is consistent with the idea that the relationship observed between Δ_age_ and BMI pointed out by the reviewer is driven by young individuals who are small and, because of their young age, tend to be overpredicted for their age due to the compression effect in the elastic net regression.

**Author response image 2. respfig2:** Results of the full male dataset are qualitatively the same as results when excluding low BMI/young males. (A,C) Chronological age in years at the time of sampling versus body mass index (kilograms/meters2) for (A) all males in our sample or (C) only males with BMI > 41 (i.e., only males above the dashed line in (A). (B,D) Results from the analysis including (B) all males in our sample (as presented in our main model) or (D) only males with BMI > 41.

Finally, we were confused about how the reviewer observed a relationship between age-adjusted BMI and Δ_age_, because we do not (using age-adjusted BMI in Supplementary Table 1, Pearson’s *r* = 0.13, p = 0.18). We could only replicate for comparison in Author response image 3 by summing our age-adjusted BMI value and the original raw BMI values to construct a new variable that we never analyzed in our study. Indeed, this approach effectively reverses the age adjustment (raw BMI versus the reviewer’s “age-adjusted BMI” in males: Pearson’s *r* = 0.94, p = 6.71 x 10^-60^).

**Author response image 3. respfig3:** (A) Age-adjusted BMI (as reported in the manuscript and provided in Supplementary Table 1) does not predict the difference between predicted age and chronological age (p=0. 18). (B) We found that Reviewer 1’s correlation, as shown in the review, can be recreated (C) only by adding raw BMI to our age-adjusted BMI values, which essentially removes the age-adjustment.

Once receiving the additional data that was not included in the original manuscript (which I asked in Minor comment 5), I was able to perform several valuations/verifications (including assays described above) to evaluate the author’s conclusion. Then I realized that the authors’ decision (statement in line 183-186) is inaccurate and has greatly affected the results. The authors should not include chronological age as a predictor in their model, even if the systematically (?) overpredicted for young and underpredicted for old animals were detected.

As we outline above, we agree that inclusion/exclusion of chronological age does greatly affect the results. However, inclusion of chronological age is in fact *essential* to producing interpretable results. This is because, due to the systematic compression effect described in lines 206-214, *any* random age-correlated variable would be spuriously associated with epigenetic age without correction for chronological age.

To illustrate this statistical confounding problem, we simulated a random variable that is correlated with chronological age to the same degree that BMI is correlated with chronological age in real data (*r* = 0.6, see code at Github (https://github.com/janderson94/BaboonEpigeneticAging/tree/master/Simulations_for_re viewers). First, we verify that this simulated variable and chronological age are significantly positively correlated at *r* = 0.6 (Author response image 4). Second, we show that this simulated variable, which has no true biological relationship with Δ_age_, nevertheless artifactually predicts Δ_age_ because of its correlation with chronological age (*r* = 0.32, p = 1.0 x 10^-4^; Author response image 4). Third, we show that this spurious relationship is eliminated by controlling for chronological age, as expected (*r* = 0.12, p = 0.16; Author response image 4).

**Author response image 4. respfig4:** (A) Our simulation (based on sampling from the multivariate normal) produces the same correlation between the simulated variable and chronological age as observed for chronological age and the BMI in the real data (r = 0.60). (B) The simulated variable significantly predicts Δage (predicted epigenetic age-chronological age) due to its correlation with chronological age. (C) Correcting for the chronological age compression effect (“residual epigenetic age”), as performed in the main manuscript, eliminates this artifactual correlation.

This spurious relationship, which occurs in the absence of chronological age correction, directly accounts for the reviewer’s findings. To demonstrate this, we repeated our simulations 1000 times to show that we consistently see a relationship with Δ_age_*before, but not after*, correction for chronological age (Author response image 5). Controlling for chronological age is therefore essential for unbiased inference, as has been previously recognized in the literature (e.g.,Hannum et al., Molecular Cell, 2013). Because this observation is an outcome of variable correlation structure and is not specific to this data set, we have not included these simulations in the revised manuscript; however, we have posted the underlying code on Github and are happy to include them at the editor’s request.

**Author response image 5. respfig5:** (A) 1000 simulations of this random variable consistently result in a negative correlation with Δage. (B) The spurious negative correlation is consistently eliminated by using residual epigenetic age. Histogram shows data from the same 1000 simulations, except with Δage corrected for the compression effect shown in main manuscript Figure 1. Histograms are colored based on the significance of the correlation between the simulated variable and either Δage or residual epigenetic age (p < 0.05 in blue, 0.05<p<0.10 in grey-blue, and p > 0.10 in grey).

By plotting real age and predicted age with their ranking information, again, I could see the epigenetic aging acceleration not only in high ranked animals (red/green dots) but also in low-ranked animals (white/blue dots) in the young to middle age range.

The reviewer is absolutely correct that, if one does not control for chronological age, young individuals will appear to have accelerated ages. This is the result of the compression effect outlined above and described in the manuscript (lines 206-214), and it is visible by eye in Figures 1A – B, which do not include the chronological age correction. As explained above, if chronological age is not taken into account, low rank (or any variable correlated with chronological age, see simulations above) will produce a spurious association with Δ_age_.

If authors included chronological age as a predictor in the model, although variations (epigenetic aging accelerations) were seen in young-middle aged animals (mostly low-ranked), the targets of assays was limited around the central range of the plotting, which is mostly high-ranked animals. That model misled the authors and affected their conclusions.

We agree that including chronological age in the model affects the results, but as outlined in detail above, it is essential (see also lines 206-214). We also now provide a new supplementary figure (Supplementary Figure S8) that shows how rank maps onto the relationship between epigenetic age and chronological age (this figure corrects for the compression effect shown in Figure 1).

I recommend that the authors use the exact same ranges on the x and y-axis and include ranking information in Figure 1 to avoid misleading the readers.

We show the x = y line on each panel, which we hope addresses this concern. However, we have retained the axis ranges in Figure 1 because we believe it is important to enable the reader to clearly discriminate the variation we are depicting rather than condensing it along the y-axis.

Then I checked the intra-individual changes. In AMB_133 and AMB_69, I could see a higher decline of epigenetic aging speed in higher rankings (rank 3 to 4, and 5 to 5, respectively) than when it changed from 4 to 14 (in AMB_133) and from 11 to 5 (in AMB_69). I also could detect the drastic decline in epigenetic aging when there was a change in rank, from 4 to 6, but none when the rank was lowered from 6 to 11 in AMB_152. I further detected that the slope of AMB_230 (2nd to 3rd time point on rank 1 to 1) was similar to AMB_198 (rank 2 to 7), showing drastic aging-speed down even though the animal kept its highest rank. In addition, the cases that showed the epigenetic aging speed slowing while keeping the same high rank (for example, AMB_230; 1 to 1, and AMB_69; 5 to 5) seems to have been omitted from the assay in Figure 3, reducing the accuracy and reliability of the assay.

All of our analyses rely on the full population sample or population subsets. We did not pursue the types of analyses the reviewer outlines because any estimate of the slope relating change in epigenetic age and change in rank will be highly inaccurate when based on connecting only two data points.

However, the comment raises an important point about how we performed our analysis for the small subset of individuals sampled three times (n = 5). For four of the five males that were sampled three times, we only included the two samples that were sampled the farthest apart in time (i.e., excluded the temporal middle sample) to maximize the age change between sample dates. For the fifth male that was sampled three times, we included the first two samples collected in time because BMI information was missing for the third sample. We made this decision to avoid including all possible pairs from the individuals sampled three times, which would place extra weight on the data from those particular individuals. We now clarify our procedure in lines 610-614.

The correlation assay gave the exact answer to the author’s hypothesis, however the results did not match their manuscript title.

Please see our responses above about the fundamental problem with analyzing Δ_age_ without controlling for chronological age.

Most importantly, as I have mentioned in Major comments, additional supportive data using other sample set/cohort/species (and/or age-BMI-range-matched cases) are essential/necessary when making a claim in a high-impact journal. Although I understand the difficulties of collecting samples, I regret to say that the manuscript with only a single sample data (without any supportive data) is not appropriate to publish in a high-impact journal, especially when the authors are trying to make a claim this impactful.

We appreciate this perspective, but respectfully disagree. Many impactful studies, especially those that require true population-based data across the life course, are conducted in single species. This has certainly been our experience and is not uncommon at *Nature Communications* or other flagship journals. For example, work on the Amboseli baboon population that sets the foundation for this paper includes Lea et al. 2018 (*PNAS*), Gesquiere et al. 2011 (*Science*), and Tung et al. 2016 (*Nature*

*Communications*), and motivating work on the consequences of nonhuman primate social status in single populations includes Silk et al. 2010 (*Current Biology*) and Snyder Mackler et al. 2016 (*Science*). Outside of primates, other work on single long-term field studies has been seminal in understanding evolution and ecology in nature (e.g., Johnston et al. 2013 *Nature* and Graham et al. 2010 *Science*, on the wild Soay sheep of St. Kilda; Grant and Grant et al. 2006 *Science*, on Darwin’s finches on Daphne Major; Huchard et al. 2016 *Nature* on the Kalahari meerkats).

I do not agree with the statement (line 191-192, 313-314) and will not be able to judge the authors claimed point unless I am provided exact evidence of ranking effects on age-related methylation using other sample sets or other species. Therefore, I have to say that the phenomenon the authors detected could just be a variation in population found in this sample group.

Please see our responses above. We note that our title explicitly refers to male wild baboons.

In addition, I found many discrepancies in the Suppl Table 1 between the original version and revised version (see orange-colored cells in attachment; xls file) and felt it was sloppy.

The reviewer is correct that there were several differences in Supplementary Table 1 between versions. During the upload process of our data to NCBI, after all analyses were complete and prior to initial submission, we mistakenly swapped several sample IDs within the subset of individuals who were sampled longitudinally (i.e., no samples were assigned to the wrong individual in the Table, but some metadata for the same individual were swapped by date). We caught this error during the first round of revisions and corrected it both in the revised Supplementary Table 1 and in the NCBI Sequence Read Archive (project accession PRJNA648767, which we removed and re-uploaded at that time). This error did not impact any of the analyses or manuscript results.

I also cannot understand why “NA”s were found in “Accuracy of birth date” of multiple sampled animals in Suppl Table 1 (see attachment), and was then deleted from the assay that used 286 samples.

Thanks for pointing this out. We had neglected to add the full metadata information for the nine samples that were added for the purposes of the longitudinal analysis (described in lines 595-597). This information has now been added to Supplementary Table 1.

I believe that the authors still can analyze data using age-range and BMI-range matched cases. I highly recommend that the authors include simple/primitive analyses, such as the ones I have provided in Suppl materials, when preparing the next version of this manuscript for submission to scientific journals.

We appreciate the time taken to revisualize and reanalyze our data. However, as our results above show, simple bivariate analyses can often be systematically misleading. Further, as outlined above and in Supplementary Table 5, removing low BMI/younger males from our data set produces qualitatively unchanged results.

Reviewer #2:I read the revised version of the manuscript, and found it much improved. Thanks to the authors for taking the comments so seriously, and for a thorough revision.I only have one comment left, which is that the structure still has issues, and is more difficult to follow than necessary. There are now 3 aims at the end of the Introduction (not numbered, but a “First” then a “Second”, then a “Finally”), but there are 4 subsections of the Results section. The language used in the Introduction to describe any of the individual aims does not map directly onto the language used for any of the Results subheadings. All of this is making it needlessly hard to track. If you have 4 results subsections, then you should have 4 aims. Easiest would be to number them, but it’s not essential. Then, use *exactly* (not similar) the same wordings to allow readers to easily track the manuscript’s aims through the Results section and onwards. For example, your first Results subsection heading is “Epigenetic clock calibration and composition”, but this is *not* how you describe your first aim in your Introduction. Why not just write in your Introduction: “Our first aim was to undertake an epigenetic clock calibration and composition”, or something similar. Set up the aims and wording structure, map it straight on to the Results section with the same wording, and it will make it easy to follow.I hope these comments are helpful. Congratulations on an excellent manuscript – James Higham

Thank you for this helpful suggestion; we’ve followed it here in the revised manuscript (lines 63-81).

Reviewer #3:The authors have done an excellent and thorough job of addressing all three sets of reviewers’ comments. I found the new analyses and discussion about BMI vs dominance rank interesting and directly relevant to the authors' findings, particularly the fact that human BMI is more variable up to unhealthy ranges in contrast to BMI in wild baboon populations. The revised manuscript presents novel findings about the epigenetic impact of rank dominance in male wild baboons. The fact that the effect is not universal is very interesting and I find the authors’ explanation of why the epigenetic aging effect of dominance rank is not found in female wild baboons to be fascinating and exactly the type of data-driven, nuanced dissection of the results needed to push the field forward. The additional analyses and more detailed explanation of the results and their implications improve an already excellent manuscript. The manuscript presents valuable new data and meticulous analyses on a unique population of wild baboons with decades of supporting data and they provide two novel results – the first epigenetic clock in a wild primate population and compelling evidence of the epigenetic aging effect of dominance rank in males – these are valuable contributions to the growing field of social and behavioral epigenetic.

Thank you for these supportive comments.